# Neurovascular coupling and bilateral connectivity during NREM and REM sleep

Kevin L Turner[1,2], Kyle W Gheres[2,3], Elizabeth A Proctor[1,2,4,5,6], Patrick J Drew[1,2,4,5]*

[1]Department of Biomedical Engineering, The Pennsylvania State University, University Park, United States; [2]Center for Neural Engineering, The Pennsylvania State University, University Park, United States; [3]Graduate Program in Molecular, Cellular, and Integrative Biosciences, The Pennsylvania State University, University Park, United States; [4]Department of Engineering Science and Mechanics, The Pennsylvania State University, University Park, United States; [5]Department of Neurosurgery, Penn State College of Medicine, Hershey, United States; [6]Department of Pharmacology, Penn State College of Medicine, Hershey, United States

**Abstract** To understand how arousal state impacts cerebral hemodynamics and neurovascular coupling, we monitored neural activity, behavior, and hemodynamic signals in un-anesthetized, head-fixed mice. Mice frequently fell asleep during imaging, and these sleep events were interspersed with periods of wake. During both NREM and REM sleep, mice showed large increases in cerebral blood volume ([HbT]) and arteriole diameter relative to the awake state, two to five times larger than those evoked by sensory stimulation. During NREM, the amplitude of bilateral low-frequency oscillations in [HbT] increased markedly, and coherency between neural activity and hemodynamic signals was higher than the awake resting and REM states. Bilateral correlations in neural activity and [HbT] were highest during NREM, and lowest in the awake state. Hemodynamic signals in the cortex are strongly modulated by arousal state, and changes during sleep are substantially larger than sensory-evoked responses.

*For correspondence:
PJD17@PSU.EDU

Competing interests: The authors declare that no competing interests exist.

## Introduction

Sleep is a ubiquitous state in animals (*Anafi et al., 2019*) that is controlled by an ensemble of nuclei and their brain-wide interactions (*Pace-Schott and Hobson, 2002*; *Sakai, 2020*; *Saper et al., 2010*). In mammals (*Cirelli, 2009*), sleep is broadly comprised of two stages: non-rapid eye movement (NREM or slow-wave sleep) and rapid eye movement (REM or paradoxical sleep) (*Weber and Dan, 2016*). Each of these states is associated with distinct patterns of electrical activity in the brain. During NREM sleep, there are broad-band increases in the local field potential (LFP) power in the cortex that are modulated at <1 Hz (*Amzica and Steriade, 1998*; *Datta and Maclean, 2007*; *Saper and Fuller, 2017*). During REM sleep, gamma band power (nominally 30–100 Hz) is elevated in the cortex (*Cantero et al., 2004*; *Le Van Quyen et al., 2010*). In the hippocampus, REM sleep is also associated with a marked increase in power in the theta band (nominally 4–10 Hz) in rodents (*Montgomery et al., 2008*; *Sullivan et al., 2014*).

While the dynamics of neural activity in the cortex and other brain structures during sleep are well characterized, the cerebrovascular manifestations of sleep are less clear. Pioneering studies using positron-emission tomography (PET) or [133]Xenon in humans have suggested that cerebral blood flow (CBF) and metabolism is reduced during NREM sleep as compared with those during awake levels, and increased above said levels during REM sleep (*Braun et al., 1997*; *Townsend et al., 1973*),

though the temporal and spatial resolutions of these techniques are poor. The degree to which CBF changes during the different sleep states appears dependent upon brain region (*Madsen et al., 1991*; *Maquet and Phillips, 1998*), complicating the interpretation of functional connectivity studies looking at correlations between brain regions where subjects may be transitioning among arousal states (*Gu et al., 2019*). Several fMRI studies have shown significant alterations in hemodynamic signals and functional connectivity during NREM sleep (*Boly et al., 2012*; *Dang-Vu et al., 2008*; *Fukunaga et al., 2006*; *Horovitz et al., 2008*; *Larson-Prior et al., 2009*; *Mitra et al., 2015*), suggesting that the blood oxygen level dependent (BOLD) signal changes during sleep. Because BOLD signals are generated by a complicated interplay of cerebral metabolism and changes in blood flow and volume (*Kim and Ogawa, 2012*), the vascular basis of these changes and their relation to neural activity are not well understood. Although functional ultrasound measures indicate cerebral blood volume rises during sleep (likely due to arterial dilation *Mateo et al., 2017*) and is correlated with hippocampal theta and gamma band power (*Bergel et al., 2018*), the relationship of cortical vascular changes to local cortical neural activity is unknown.

Understanding the vascular basis of hemodynamic signals during sleep is relevant to many aspects of brain health and function. First, BOLD signal changes during sleep are associated with movement of cerebrospinal fluid (CSF) (*Fultz et al., 2019*), and the movement of CSF is thought to play an important role in maintaining brain health (*Simon and Iliff, 2016*; *Tarasoff-Conway et al., 2015*; *Xie et al., 2013*). Elucidating the vascular changes associated with these fluid movements would help resolve the actual drivers of fluid movement. Secondly, there is accumulating evidence that arousal state transitions drive large hemodynamic changes, both in animals performing tasks (*Cardoso et al., 2019*), and in humans and animals undergoing 'resting-state' studies (*Chang et al., 2016*; *Liu, 2016*; *Liu et al., 2018*; *Tagliazucchi and Laufs, 2014*). If the hemodynamic signals during the periods of altered arousal are large or correlated enough, the activity during the sleep states could dominate the functional connectivity signal. Complicating mechanistic studies in mice is the fact that head-fixed mice do not close their eyes during NREM and REM sleep (*Yüzgeç et al., 2018*), meaning that without careful monitoring or a task it is possible that many studies examining 'resting-state' correlations in head-fixed mice were compounded by sleep.

Here we measured behavior, neural activity, blood volume and arteriole dilations from head-fixed mice during the awake state and NREM and REM sleep. We found that arteriole dilations and blood volume changes during NREM and REM sleep could be two to five times larger than those occurring in the awake animal. The correlations between neural activity and hemodynamic signals was greatly increased during NREM sleep, and the functional connectivity between interhemispheric regions of somatosensory cortex also increased.

## Results

We used intrinsic optical signal (IOS) (*Huo et al., 2014*; *Winder et al., 2017*) (14 mice, nine males) and 2-photon microscopy (*Drew et al., 2011*; *Echagarruga et al., 2020*; *Shih et al., 2012a*) (six mice, two males) in concert with electrophysiology to measure neural activity (*Buzsáki et al., 2012*; *Harris et al., 2016*) from the whisker representation of somatosensory cortex and the CA1 region of the hippocampus in un-anesthetized, head-fixed C57BL6/J mice (*Figure 1A*) during the light cycle. After mice were habituated to head-fixation, data was acquired from each mouse for 5–7 days. We obtained 357.2 total hours of data from these mice (mean: 23 ± 5.1 hr per mouse from IOS mice *Supplementary file 1*), 5.8 ± 2.0 hr per mouse from 2-photon imaged mice (*Supplementary file 1*). All experiments were performed during the animal's light cycle. We tracked whisker position (*Kleinfeld and Deschênes, 2011*; *O'Connor et al., 2010*), body movement, and nuchal muscle EMG (*Datta and Maclean, 2007*; *Veasey et al., 2000*), as spontaneous 'fidgeting' behaviors drive a substantial portion of neural activity and hemodynamic signals in the awake mouse (*Drew et al., 2019*; *Musall et al., 2019*; *Stringer et al., 2019*; *Winder et al., 2017*), and these measures can be used to determine the arousal state of the animal. We recorded neural activity differentially across stereotrodes from the whisker representation of the somatosensory cortex and the hippocampus to reject non-local electrical signals (*Buzsáki et al., 2012*; *Kajikawa and Schroeder, 2011*; *Nicholson and Freeman, 1975*). For IOS data, we used a bootstrap aggregation random forest to determine the arousal state from these behavioral measures and hippocampal and cortical LFPs (see Materials and methods), categorizing every non-overlapping five second interval into one of three

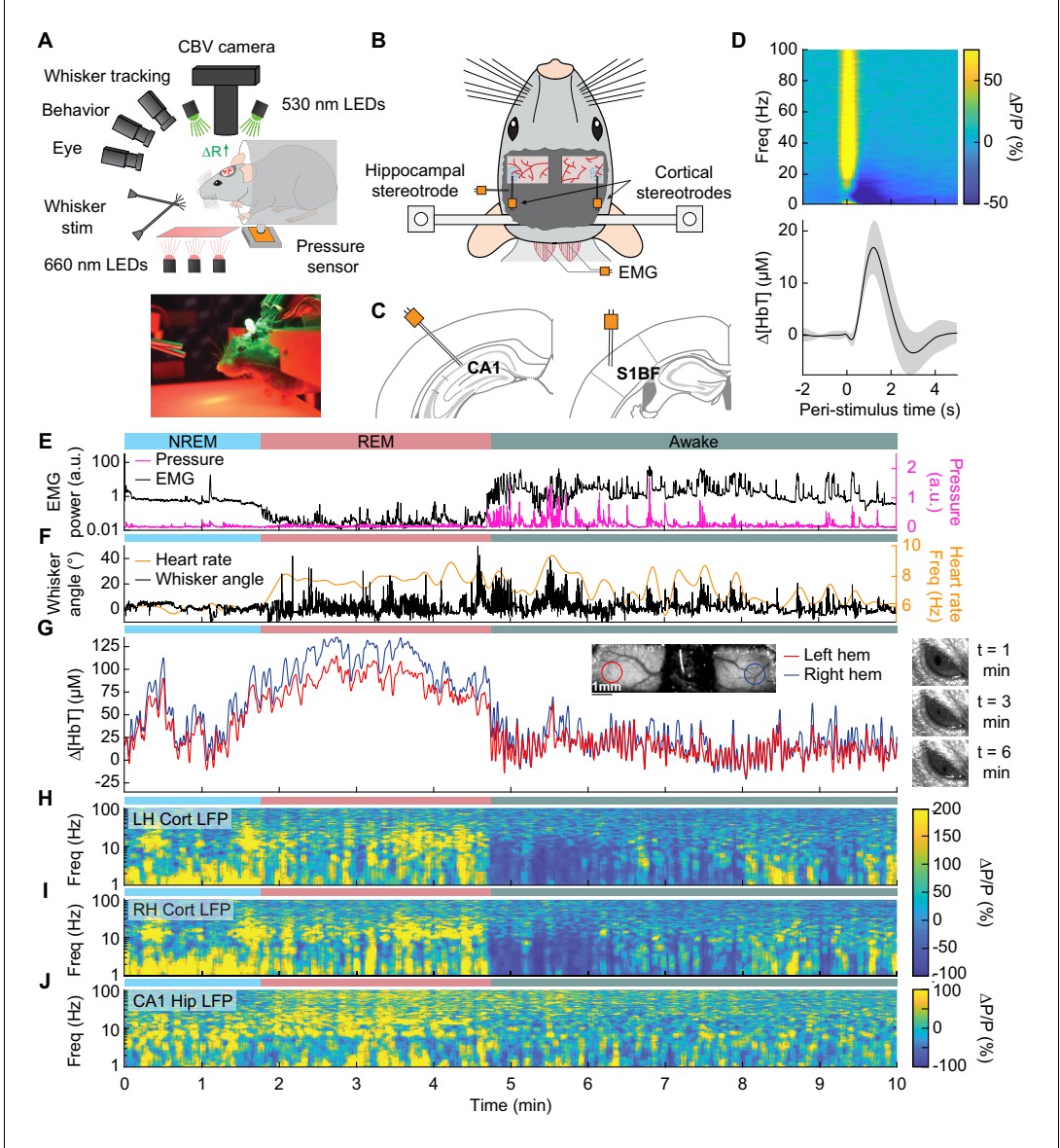

**Figure 1.** Sleep drives large changes in cerebral blood volume. (A) Schematic of IOS experimental setup. The brain is illuminated with 530 nm LEDs, and changes in reflected light captured by a CCD camera mounted above the head. Other cameras track the whiskers (illuminated by 660 nm LEDs beneath the animal), the eye (illuminated by 780 nm LEDs), and changes in animal behavior. A piezo sensor to record changes in body motion is located beneath the animal, which rests head-fixed in a cylindrical tube. Tubes direct air to the distal part of the whiskers (but not the face), and do not interfere with volitional whisking. (B) Schematic showing the locations of the bilateral thinned-skull windows and recording electrodes. Each electrode consists of two Teflon-coated tungsten wires (~100 μm tip spacing), while the EMG electrode consists of two stainless-steel wires with several mm of insulation stripped off each end, inserted into adjacent nuchal muscles. (C) Left: Diagram showing hippocampal CA1 recording site. Right: Diagram of somatosensory cortex recording site. Adapted from Figure (52) (left) and Figure (42) (right) of The Mouse Brain in Stereotactic Coordinates, 3rd Edition (*Franklin and Paxinos, 2007*). (D) Average neural and hemodynamic responses to contralateral whisker stimulation (n = 14 mice, 28 hemispheres, 110 ± 70 stimulations per animal). Top: average normalized change in LFP power (ΔP/P) in the somatosensory cortex in response to contralateral whisker stimulation. Bottom: mean change in total hemoglobin (Δ[HbT]) within the ROI. Shaded regions indicate ± 1 standard deviation. (E-J) Example showing the hemodynamic and neural changes accompanying transitions among the NREM, REM and awake states. (E) Plot of nuchal muscle EMG power and body motion via a pressure sensor located beneath the mouse. (F) Plot of the whisker position and heart rate (G) Changes in total hemoglobin Δ[HbT] within the ROIs. Inset shows images of the two windows and respective ROIs. (H,I) Normalized left and right vibrissae cortex LFP power (ΔP/P). (J) Normalized CA1 LFP power.

The online version of this article includes the following figure supplement(s) for figure 1:

**Figure supplement 1.** Localization of electrodes and hemodynamic regions of interest.

**Figure supplement 2.** Whisker stimulation causes increases in neural activity and blood volume.

*Figure 1 continued on next page*

categories: rfc-Awake, rfc-NREM, or rfc-REM, where 'rfc' denotes the arousal state assigned via an automated random forest classifier. rfc-Awake periods were further characterized into *awake rest* (defined as periods lasting longer than 10 s that lack whisker and body movement), *awake whisking* (defined as bouts of whisking lasting between 2 and 5 s in duration), and *awake stimulation* (directed air puffs to contralateral whiskers). Each of these three awake behaviors were manually identified in rfc-Awake data to exclude transitional arousal states. Periods of *contiguous NREM* sleep and *contiguous REM* sleep were defined as events lasting at least 30 s and 60 s, respectively (*Supplementary file 1*). The 2-photon microscopy data was manually scored in a fashion similar to rfc-IOS data (Manual(m)-Awake, m-NREM, m-REM) and then further subdivided into awake rest, awake whisking, contiguous NREM, and contiguous REM (no whisker was stimulation presented during 2-photon experiments) (*Supplementary file 1*). As awake behaviors can be much shorter in duration than NREM and REM events, we used different durations when categorizing each arousal state. We chose the specific durations for each arousal state based on the typical minimum duration of each event, allowing us to catch the neural and vascular dynamics of each behavior. All reported values are mean ± standard deviation, unless otherwise indicated.

## Sleep drives larger fluctuations than awake behaviors

We first examined how arousal state affected hemodynamic signals using intrinsic optical signal imaging (*Huo et al., 2014*; *Sirotin and Das, 2009*; *Vazquez et al., 2014*), which detects changes in total hemoglobin [HbT] from changes in reflectance. Periods of awake rest without whisking or stimulation were set as the zero baseline (see Methods). Increases in blood volume (vasodilation) cause decreases in reflectance, which are converted into hemoglobin changes (Δ[HbT]) using the Beer-Lambert law (*Ma et al., 2016*). These hemodynamic measurements were done bilaterally through a polished and reinforced thinned-skull window (*Figure 1A*; *Drew et al., 2010a*; *Shih et al., 2012b*) encompassing the whisker-related section of the somatosensory cortex (*Petersen, 2007*). The LFP was recorded from the vibrissa cortex within the window (*Figure 1B,C*) to provide a direct measurement of neural activity from whisker-related portions of somatosensory cortex from both hemispheres (*Winder et al., 2017*). To assist in arousal state classification, we also recorded the CA1 hippocampal LFP (*Figure 1B,C*), nuchal muscle electromyography (EMG), motion of the whiskers, body motion, and heart rate. Hemodynamic measurements were taken from a region of interest within a 1 mm diameter circle centered on the pixels that showed the highest cross-correlation between reflectance and gamma band power (1–2 s lag) (*Winder et al., 2017*) during the first 15–60 min of data (see *Figure 1—figure supplement 1A–C*). This period contains not just rest, but also whisking and some whisker stimulation. This differs from *Winder et al., 2017*, where only resting periods, lack whisking and stimulation, were used to define the ROI. These pixels corresponded to a region putatively within the vibrissa cortex and were consistent across imaging days. This is consistent with the hemodynamic point-spread function having full-width at half-max of several hundred microns (*Vazquez et al., 2014*) and the hyperemic response being conducted over several hundred microns (*Rungta et al., 2018*). The spatial scale of the measured hemodynamic signal is very close to that measured by the electrodes.

At the beginning of each day's imaging session, we stimulated the vibrissae on either side with a brief puff of air, which drove canonical neural and vascular responses (*Figure 1D*). Consistent with previous work (*Winder et al., 2017*), gamma band power in the somatosensory cortex (30–100 Hz) increased by 78.1 ± 66.7%, followed by a 16.8 ± 5.1 µM increase in [HbT] (corresponding to a −2.4 ± 0.7% decrease in reflectance, ΔR/R, see *Figure 1—figure supplement 2*). Volitional, awake whisking led to an increase in gamma band power and [HbT] that increased with the duration of the whisking

event. Brief (0.5–2 s), moderate (2–5 s), and extended (>5 s) whisking events lead to an increase of 5 ± 4.8%, 9.7 ± 7%, and 16.5 ± 10.8% in gamma band power in the somatosensory cortex. These changes led to an increase in [HbT] of 2.5 ± 1.4 μM, 7.1 ± 3.3 μM, 12.1 ± 5.6 μM respectively (see *Figure 1—figure supplement 3*).

In comparison to the sensory-evoked responses, we found much larger changes in [HbT] associated with sleep. NREM sleep is characterized by low EMG activity, lack of whisker and body movement, and pronounced power in the low-frequency bands of the cortical LFP (*Figure 1E–J*, *Videos 1–3*; *Scammell et al., 2017*; *Vyazovskiy and Harris, 2013*). REM sleep is characterized by neck muscle atonia, sporadic whisker movement, and increased theta band power in the hippocampus (*Figure 1E–J*; *Hobson and Pace-Schott, 2002*; *Walker and Stickgold, 2004*). During contiguous NREM sleep, we observed large oscillations in total hemoglobin, up to 87.3 ± 9.9 μM [HbT] in peak-to-peak amplitude, compared to 32.3 ± 4.4 μM during awake rest (generalized linear mixed-effects (GLME), p<9 × 10$^{-32}$). During contiguous REM bouts, there was a prolonged (>30 s) increase of [HbT] of up to 107.7 ± 13.3 μM, greatly surpassing the maximum of 17 ± 3.2 μM during awake rest (GLME, p<3.5 × 10$^{-55}$, see *Figure 1—figure supplement 4A,B*). Examples of sleep-related changes in blood volume are shown in *Figure 1E–J* and *Figure 1—figure supplement 5*, *Figure 1—figure supplement 6*, *Figure 1—figure supplement 7*, and *Figure 1—figure supplement 8*. Note that as previously observed (*Yüzgeç et al., 2018*), the eyes of the mouse are open during both REM and NREM sleep states (*Figure 1G*). These results show that sleep in head-fixed mice is associated with large increase in blood volume, particularly during REM sleep.

## Mice regularly enter into NREM and REM sleep during head-fixation

We then asked how often head-fixed mice sleep, and how the probability of sleep changes over time from the start of head-fixation. A hypnogram for a single mouse over 6 days with 5 s resolution is shown in *Figure 2A*. The white lines denote brief breaks in recording while data is saved. From this example, it is clear that the mouse has many periods of NREM and REM sleep interspersed with awake. Periods of REM sleep canonically follow NREM sleep (*Saper et al., 2010*). REM periods are typically followed by awakening, though REM sleep can be followed by NREM periods. The mean percentage of each classified state is shown in *Figure 2B*, and the breakdown for individual animals in *Figure 2C*. We noted no clear difference in (rfc-)Awake:NREM:REM sleep ratios between males and females, so they were pooled for other analyses. Plotting the probability of finding the mouse in each of the states as a function of time throughout the imaging session (*Figure 2D*) shows that as time goes by the mouse is more likely to be

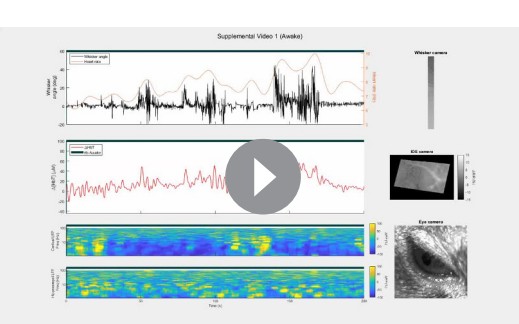

**Video 1.** Arousal and associated changes in neural activity and hemodynamics in the awake state. Whisker motion, IOS reflectance, and eye camera activity are shown alongside measurements of neural activity and hemodynamics. The awake state shows a large amount of whisker motion and elevations in heart rate. The eye is open. High-frequency neural activity increases and low-frequency activity decreases during whisking events, with corresponding decreases in reflectance. An increase in blood volume/[HbT] corresponds to a decrease in pixel reflectance, as more light is absorbed by the increase in hemoglobin.
https://elifesciences.org/articles/62071#video1

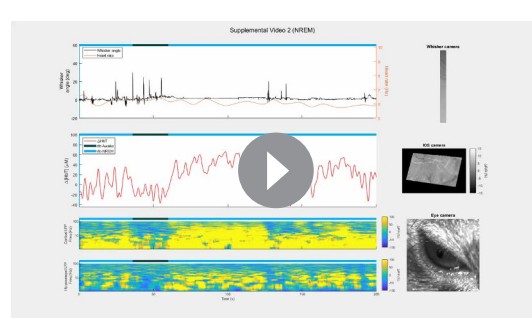

**Video 2.** Arousal and associated changes in neural activity and hemodynamics during NREM sleep. Whisker motion, IOS reflectance, and eye camera activity are shown alongside measurements of neural activity and hemodynamics. The NREM state shows little whisker motion and a lower heart rate. The eye is still open. Low-frequency (delta band) cortical neural activity is elevated, and there are large changes in reflectance.
https://elifesciences.org/articles/62071#video2

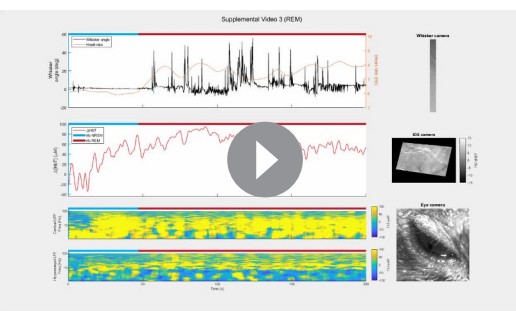

**Video 3.** Arousal and associated changes in neural activity and hemodynamics during REM sleep. Whisker motion, IOS reflectance, and eye camera activity are shown alongside measurements of neural activity and hemodynamics. The transition into the REM state shows an increase in whisker motion and an increase heart rate, similar to the awake state. The eye remains open. The neural activity in the hippocampal theta band and upper frequencies of cortical neural activity increase while [HbT] increases substantially.
https://elifesciences.org/articles/62071#video3

asleep. As awake mice typically whisk every ~10 s, we quantified the probability that the mouse had fallen asleep after a given period lacking whisking and body movement ('Rest') across all animals. We found that during only ~50% of 'resting' events lasting 10–15 s were the mice awake for the whole event, and with longer 'resting' events showing even lower probability of wakefulness (*Figure 2E*). This result is reminiscent of studies in humans showing the probability of being awake falls rapidly with time during a resting-state fMRI scan (*Tagliazucchi and Laufs, 2014*), though human sleep/wake behavior is much less fragmented than in the mouse, and the transition times are correspondingly longer. EMG activity (*Figure 2F*) was much lower during sleep than the awake state (see *Figure 2—figure supplement 1A*: rfc-NREM: $p<3.2 \times 10^{-12}$, rfc-REM: $p<1.7 \times 10^{-21}$, GLME), though the amount of whisking (quantified as the variance in the whisker angle) was much smaller during rfc-REM and rfc-NREM sleep than during the awake state (*Figure 2G*, see *Figure 2—figure supplement 1B*: rfc-NREM: $p<1.3 \times 10^{-12}$, rfc-REM: $p<0.003$, GLME). The heart rate was quantified by finding the peak in the intrinsic signal power spectrum between 5–15 Hz (*Huo et al., 2015a*; *Huo et al., 2015b*; *Winder et al., 2017*). Heart rate was lowest during rfc-NREM sleep (*Figure 2H*, see *Figure 2—figure supplement 1C*: rfc-NREM: $p<1.8 \times 10^{-13}$, rfc-REM: $p<0.004$, GLME), though heart rate during contiguous REM was comparable to heart rate in the awake, resting mouse, and the heart rate was elevated during awake whisking (*Figure 2I*, Awake Whisk: $p<1.1 \times 10^{-10}$, contiguous NREM: $p<3.8 \times 10^{-14}$, contiguous REM: $p<0.0009$, GLME). These observations demonstrate the prevalence of sleep in 'resting-state' data in head-fixed mice and show how unimodal measures, such as whisker movement or heart rate alone, are insufficient to detect these sleep states.

## Sleep drives arteriole dilations much larger than those seen in the awake brain

The intrinsic optical signal contains contributions from arteries, veins, and capillaries (*Huo et al., 2015a*; *Huo et al., 2015b*; *Zhang et al., 2019*), so to better understand how arterioles changes contribute to the signal, we used two-photon microscopy (*Shih et al., 2012a*) to image pial and penetrating arterioles (25 and 4 arterioles respectively, from six mice) (*Figure 3A*). We recorded hippocampal and cortical LFP contralateral to the imaging window (*Figure 3B,C*). Two-photon imaging in awake mice has shown that sensory stimulation and locomotion drive dilations of arteries of approximately 20% above the baseline diameter (*Drew et al., 2011*; *Echagarruga et al., 2020*; *Gao and Drew, 2016*; *Huo et al., 2015a*; *Huo et al., 2015b*) and whisking drives dilations of 5–10% (*Drew et al., 2020*). We observed a similar dilation with spontaneous whisking as a function of whisking duration. Brief (0.5–2 s), moderate (2–5 s), and extended (>5 s) whisking events led to an increase in vessel diameter of 0.8 ± 0.9%, 8.3 ± 4.9%, and 10.9 ± 4.3% respectively (*Figure 3D*, see *Figure 3—figure supplement 1*). Examples of sleep-related changes in arteriole diameter is shown in *Figure 3E–I* and *Figure 3—figure supplement 2*, *Figure 3—figure supplement 3*, *Figure 3—figure supplement 4*, and *Figure 3—figure supplement 5*. In comparison to the awake state, arteriole diameter during NREM sleep follows a low-frequency dilation/constriction with peak dilations that can exceed those seen during moderate whisking (*Figure 3D*). During REM sleep, the arterioles slowly dilate over tens of seconds, and can reach peak dilations in excess of 50% of the baseline diameter during periods of awake rest. The dilation amplitudes during NREM and REM sleep dwarf those seen in the awake animals. Penetrating arterioles had similar sleep-wake dynamics as the pial arterioles (see *Figure 3—figure supplement 2* vs *Figure 3—figure supplement 3*) and were

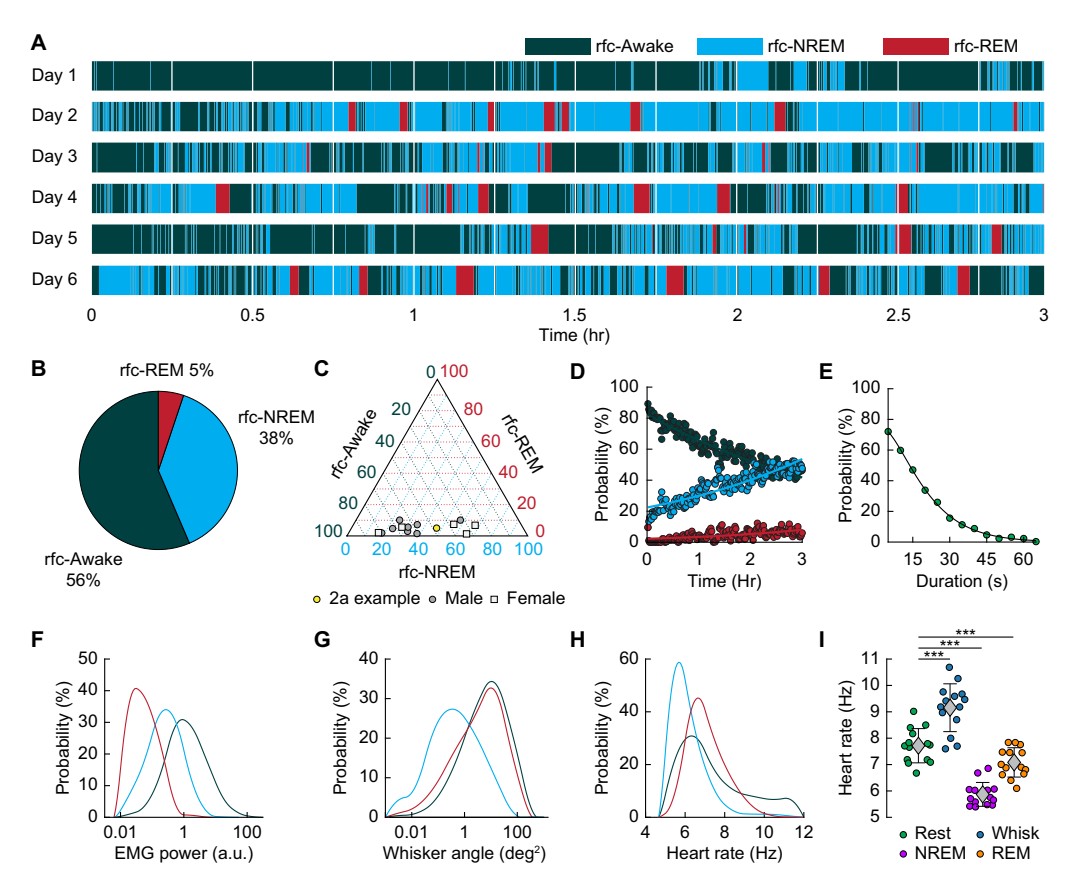

**Figure 2.** Mice rapidly and repeatedly transition between wake and sleep during head-fixation. (A) Hypnogram showing the arousal states for a single mouse over six days. The hypnogram has a resolution of 5 s. White denotes breaks in data acquisition for saving of data. Note the rapid and frequent transitions between rfc-Awake, rfc-NREM and rfc-REM. (B-I) n = 14 mice. (B) Average percentage of the time spent in each arousal state. (C) Ternary plot showing each individual animal's percentage in each arousal state. (D) Average probability of an animal being classified in a given arousal state as a function of time since the start of the session. Mice are progressively more likely to sleep and to be in REM sleep the longer they have been head-fixed E, Average probability of the animal being awake as a function of the duration of the period without movement. Mice are more likely to be asleep the longer they go without moving their whiskers or body. (F) Probability distribution of the mean EMG power during individual arousal states (5 s resolution) taken from all animals. (G) Probability distribution of variance in the whisker angle during individual arousal states (5 s resolution) taken from all animals. (H) Probability distribution of the mean heart rate for each arousal state. (I) Mean heart rate during different arousal states. Circles represent individual mice and diamonds represent population averages ± 1 standard deviation. *p<0.05, **p<0.01, ***p<0.001 GLME.

The online version of this article includes the following figure supplement(s) for figure 2:

**Figure supplement 1.** Behavioral measurements demarcate transitions between arousal states.

**Figure supplement 2.** Random forest model validation.

combined into a single group for all analysis. These results show that both NREM and REM sleep cause pronounced cortical arteriole dilations. Contiguous NREM was associated with peak-to-peak diameter changes of 38 ± 15.8%, compared to 16.6 ± 4% seen during awake rest (GLME, $p<3.6 \times 10^{-10}$). Contiguous REM drove a prolonged ramp-up to peak dilation of 49.9 ± 9.1% compared to a peak dilation of 7.4 ± 2.6% during the awake resting baseline (GLME, $p<2 \times 10^{-24}$, see *Figure 1—figure supplement 4C,D*). These large dilations were followed by a pronounced and rapid constriction upon waking.

## Hemodynamic changes associated with transitions between arousal states

To quantify the dynamics of sleep-related changes in blood volume and arteriole diameter, we looked at the dynamics of these signals, as well as LFP and EMG signals, aligned to the arousal state

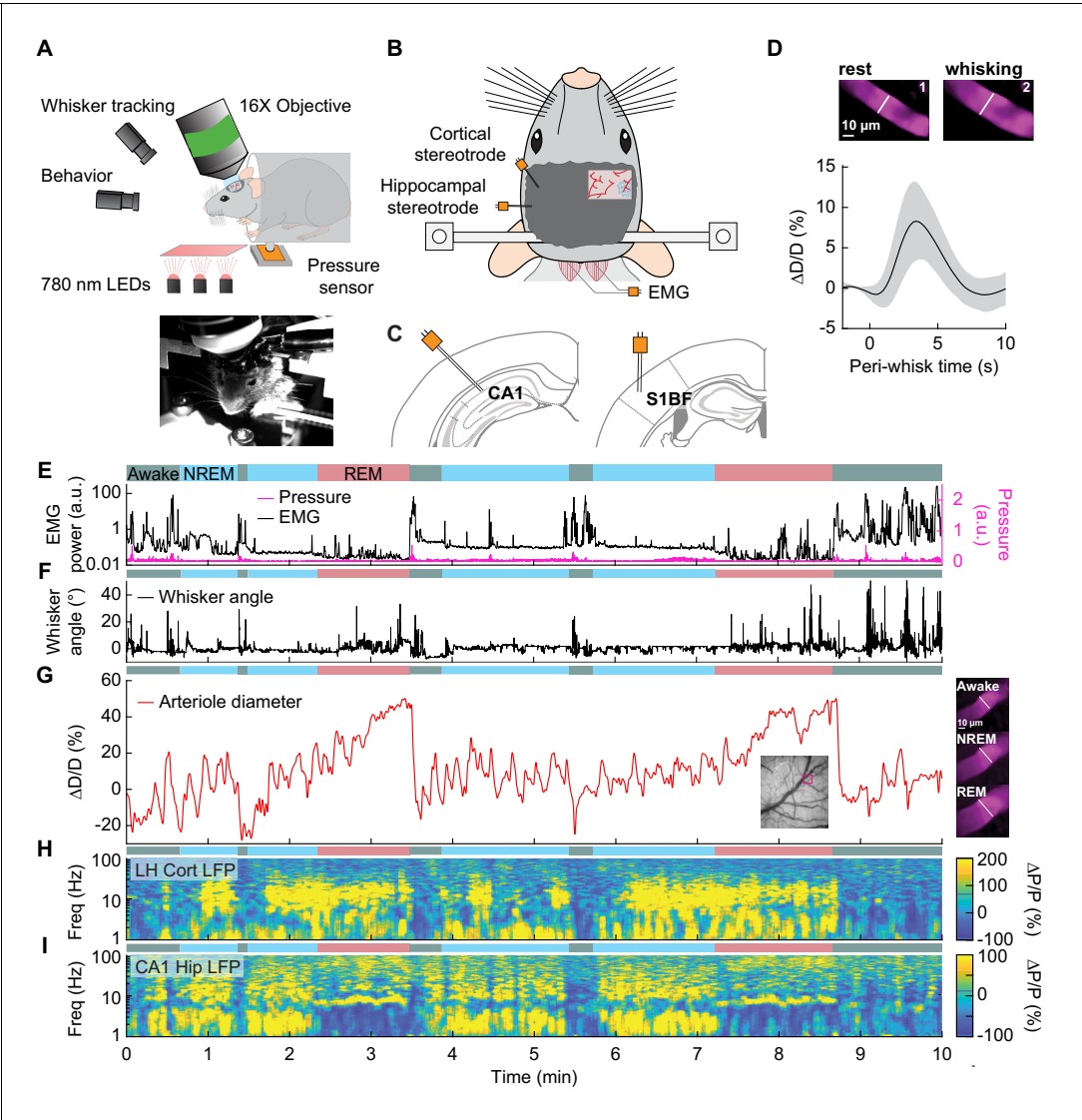

**Figure 3.** Sleep drives arteriole dilatations larger than those seen in the awake brain. (A) Schematic of two-photon experimental setup. (B) Schematic of thinned-skull window and electrode recording sites. (C) Left: Diagram showing hippocampal CA1 recording site. Right: Diagram of vibrissae cortex recording site. Adapted from Figure (52) (left) and Figure (42) (right) of The Mouse Brain in Stereotactic Coordinates, 3rd Edition (*Franklin and Paxinos, 2007*). (D) Average response to awake volitional whisking (n = 6 mice, 29 arterioles). Top: Example showing a single arteriole's diameter during rest and during a brief whisking event. Bottom: average change in arteriole diameter ΔD/D (%) during brief (2–5 s long) whisking events. Shaded regions indicate ±1 standard deviation. (E-I) Example showing the vascular and neural changes accompanying transitions among the NREM, REM and awake states. (E) Nuchal muscle activity through normalized EMG and body motion via a pressure sensor located beneath the mouse. (F) Whisker position. (G) Changes in arteriole diameter ΔD/D (%). (H) Normalized vibrissae cortical LFP. (I) Normalized CA1 LFP power.

The online version of this article includes the following figure supplement(s) for figure 3:

**Figure supplement 1.** | Volitional whisking causes arteriole dilation.
**Figure supplement 2.** Arteriole dilatations during sleep are larger than those during the awake state.
**Figure supplement 3.** Arteriole dilatations during sleep are larger than those during the awake state.
**Figure supplement 4.** Arteriole dilatations during sleep are larger than those during the awake state.
**Figure supplement 5.** Arteriole dilatations during sleep are larger than those during the awake state.

transition (*Figure 4A*). We used transitions between two arousal states, where time in each arousal state was at least 30 s in duration. Mammals will typically progress through the Awake-NREM-REM-Awake pattern of the sleep cycle (*Saper et al., 2010*), even if the awake periods between the end of a REM event and the initiation of the subsequent NREM period are very brief. The transition from

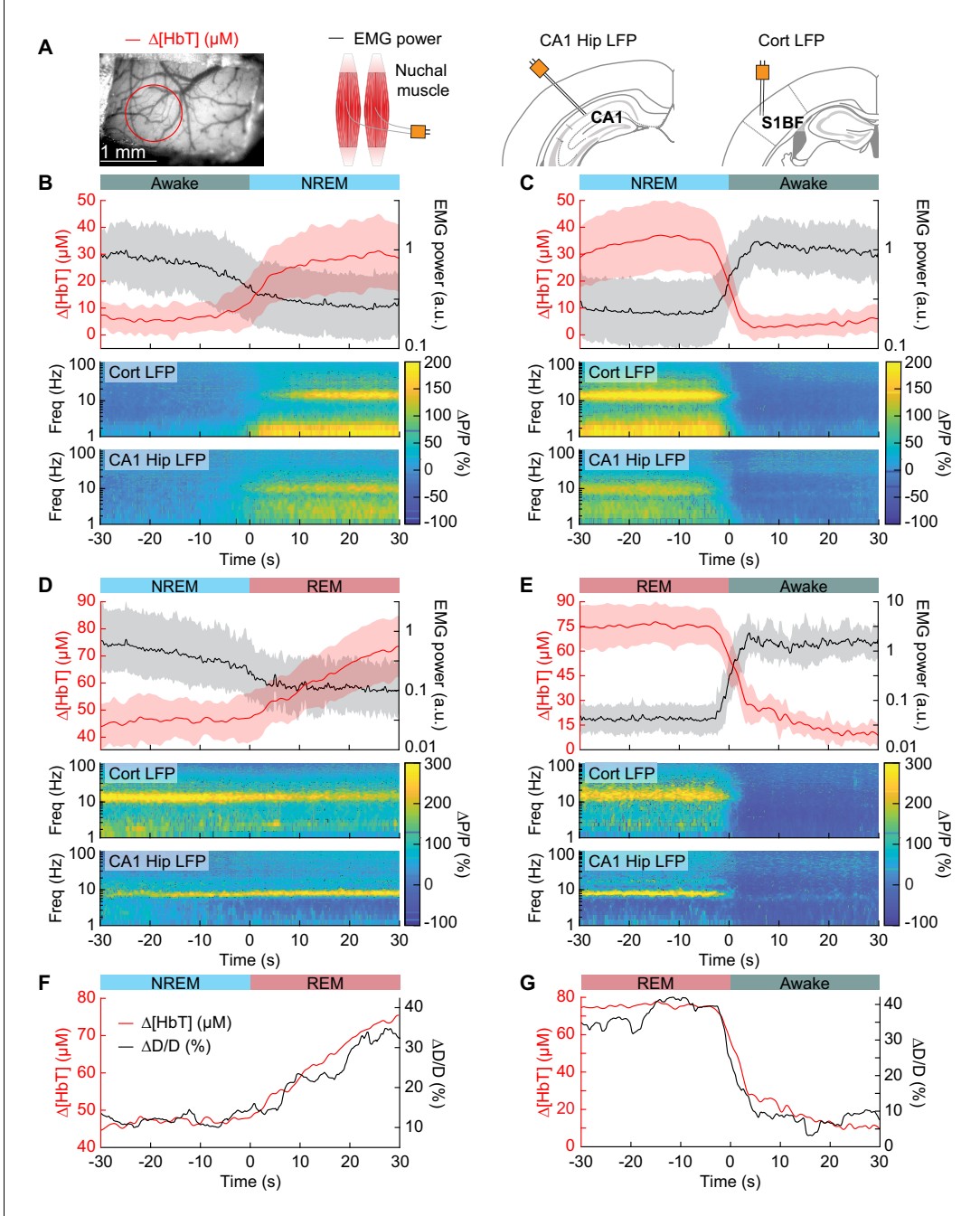

**Figure 4.** Vasodilation tracks transitions between arousal states. (A) Schematic of each data type. (B) Transition from periods classified as rfc-Awake into periods classified as rfc-NREM. Top: Average change in total hemoglobin Δ[HbT] within the ROI and normalized change in EMG power. Shaded regions indicate ± 1 standard deviation (n = 14 mice, 28 hemispheres). (C) Transition from periods classified as rfc-NREM into periods classified as rfc-Awake. (D) Transition from periods classified as rfc-NREM into periods classified as rfc-REM. (E) Transitions from periods classified as rfc-REM into periods classified as rfc-Awake. (F) Mean arteriole diameter during the transition of NREM into REM (n = 5 mice, 8 arterioles). (G) Mean arteriole diameter during the transition from REM into Awake (n = 5 mice, 8 arterioles). Note that the EMG scales are different across conditions. The online version of this article includes the following figure supplement(s) for figure 4:

**Figure supplement 1.** Transitional changes in hemodynamics are consistent across each day.

the rfc-Awake state into rfc-NREM (*Figure 4B*) shows an increase in total hemoglobin from the baseline of very low [HbT] in the awake state, up to 30 μM over the course of 30 s. During this time, the EMG power decreases with similar temporal dynamics. The LFP power in the vibrissa cortex shows increased power in the delta band [1–4 Hz] of around 300%. The transition from rfc-NREM into the rfc-Awake state (*Figure 4C*) was largely a temporally reversed version of the awake to NREM transition, where [HbT], LFP, and EMG signals all quickly return to baseline values as the animal wakes up. During the transition from rfc-NREM to rfc-REM (*Figure 4D*) there was a slow increase in total hemoglobin from 45 μM up to 75 μM. During the NREM-REM transition, the muscles become atonic and the EMG power decreased even more. The theta band [4–10 Hz] power in the hippocampal LFP increased by approximately 300%. The transition from rfc-REM into the rfc-Awake state (*Figure 4E*) was the largest in terms of the magnitude of the total hemoglobin change, as the large blood volume increase seen during REM rapidly reverses (within seconds) as the animal wakes up. Transitions from awake to REM, as well as REM to NREM are possible, but are much less common, and did not occur often enough to quantify reliably. The amplitude and temporal dynamics of Δ[HbT] transitions between each arousal state were consistent across imaging days (see *Figure 4—figure supplement 1*).

When looking at single arterioles during the transition from NREM to REM, arterioles will on average go from an approximate 20% dilation, up to approximately 40% dilation after a minute or so in REM (*Figure 4F*). Upon waking from REM, arterioles constrict back to the baseline diameter within a few seconds (*Figure 4G*). When the arterial diameter and Δ[HbT] are plotted together, they show very similar temporal dynamics during the NREM to REM transition and during the REM to awake transition (*Figure 4F,G*). The close match in dynamics suggests that arteriole dilations are a significant driver of the blood volume changes during sleep, as is seen in the awake brain (*Huo et al., 2015a*; *Rungta et al., 2018*).

## Cortical hemodynamic signals increase during NREM and REM sleep

We quantitatively compared hemodynamic signals during different arousal states. For this quantification, we used awake resting events $\geq$ 10 s in duration, awake whisking events 2–5 s in duration, brief stimulations of the whiskers, contiguous NREM sleep events $\geq$ 30 s in duration, and contiguous REM sleep events $\geq$ 60 s in duration. The average Δ[HbT] during each day's awake resting condition was set as zero (n = 14 mice, 28 hemispheres). Awake whisking events between 2 and 5 s in duration caused a slight increase in [HbT] of 3.1 ± 2.5 μM (GLME, p<0.11). The increase caused by whisking and fidgeting, which are the primary drivers of resting-state neural and hemodynamic signals in the awake mouse (*Drew et al., 2019*; *Drew et al., 2020*; *Musall et al., 2019*; *Stringer et al., 2019*; *Winder et al., 2017*), were dwarfed by those that occurred during contiguous NREM sleep (32.2 ± 11.2 μM, GLME, $p < 1 \times 10^{-34}$) and during contiguous REM sleep (77.1 ± 11.9 μM, GLME, $p < 1 \times 10^{-76}$). These sleep-driven changes were much larger than sensory-evoked changes generated by contralateral whisker stimulation (12.9 ± 5.6 μM, GLME, $p < 4.9 \times 10^{-10}$). The probability distribution of Δ[HbT] during each arousal state is shown in *Figure 5D*. There was a clear and pronounced separation in the hemodynamic signals between the sleep and awake states, as well as between contiguous NREM and REM sleep. However, the increases in [HbT] during sleep were smaller than those caused by the vasodilator isoflurane (*Flynn et al., 1992*; *Gao et al., 2015*; *Figure 5—figure supplement 1*), showing the vessels are not maximally dilated during sleep.

The average dilation of arterioles during each arousal state followed the same trend as the Δ[HbT] data (*Figure 5B*). Awake whisking events spurred an average diameter increase of 4.2 ± 3.2% (GLME, p<0.0006, n = 6 mice, 29 arterioles), with contiguous NREM sleep dilating 9.4 ± 9.3% (GLME, $p < 8.7 \times 10^{-19}$, n = 6 mice, 21 arterioles) and contiguous REM sleep reaching dilations of 32.7 ± 8.6% (GLME, $p < 5 \times 10^{-34}$, n = 5 mice, 10 arterioles). The probability distribution of arteriole ΔD/D during each arousal state with data taken from all animal's events is shown in *Figure 5E* Recordings of changes in blood flow measured using laser Doppler flowmetry (ΔQ/Q) followed the same trend (*Figure 5C*) (n = 8 mice, one mouse was excluded due to a weak signal), with the largest flow increases during sleep. Awake volitional whisking events caused negligible increases of 0.4 ± 2.6% (GLME, p<0.91). Flow changes were more prominent during contiguous NREM sleep, 14.4 ± 9.1% (GLME, p<0.0004), and even more-so during contiguous REM sleep, 28.3 ± 12.8% (GLME, $p < 1.5 \times 10^{-8}$). The probability distribution of ΔQ/Q during each arousal state with data taken from all animal's events is shown in *Figure 5F*. These measurements of blood volume, arteriole diameter,

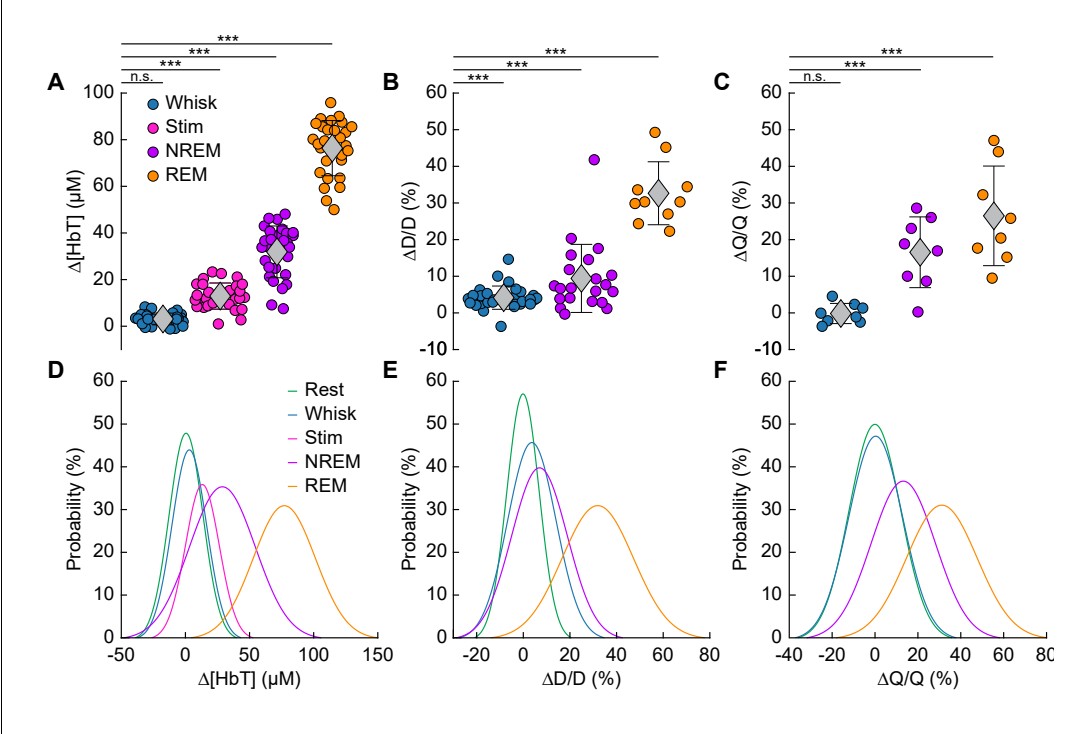

**Figure 5.** Increases in blood volume and arterial diameter during NREM and REM sleep. (**A**) Average change in total hemoglobin Δ[HbT] within the ROI. Circles represent individual hemispheres of each mouse, diamonds represent population averages, with error bar showing ±1 standard deviation (n = 14 mice, 28 hemispheres). (**B**) Average change in peak arteriole diameter ΔD/D (%) (n = 6 mice, 29 arterioles for whisking; n = 6 mice, 21 arterioles for contiguous NREM; n = 5 mice, 10 arterioles for contiguous REM). (**C**) Average change in volumetric flux (ΔQ/Q, %) measured with laser Doppler flowmetry during different arousal states (n = 8 mice). (**D**) Probability distribution of Δ[HbT] during each arousal state. (**E**) Probability distribution of ΔD/D (%) during each arousal state. (**F**) Probability distribution of ΔQ/Q (%) during each arousal state. *p<0.05, **p<0.01, ***p<0.001 GLME.

The online version of this article includes the following figure supplement(s) for figure 5:

**Figure supplement 1.** Isoflurane drives larger vasodilations than sleep.

and flow all show a consistent and pronounced trend of vasodilation in the somatosensory cortex during sleep which far surpassed those seen during volitional behaviors and sensory-evoked stimulation in the awake animal.

## Neurovascular coupling is strongest during NREM sleep

It has previously been shown that neurovascular coupling is similar across awake arousal states (*Winder et al., 2017*) and that both spontaneous and sensory-evoked hemodynamics are most strongly correlated with gamma band power and multi-unit average (MUA – a measure of local spiking activity) in the absence of overt stimulation (*Mateo et al., 2017*; *Schölvinck et al., 2010*; *Shmuel and Leopold, 2008*; *Winder et al., 2017*). To see if these relationships held true for neurovascular coupling in other arousal states, we looked at the relationship between the power in different frequency bands of the LFP or multi-unit activity and Δ[HbT] during periods of contiguous NREM and REM sleep. Correlations between the MUA and Δ[HbT] as well as the LFP and Δ[HbT] during awake rest showed a peak correlation of 0.24 ± 0.06 for the MUA and 0.16 ± 0.04 for the gamma band, consistent with previous work (*Winder et al., 2017*; *Mateo et al., 2017*). The hemodynamic response lagged the MUA by 1.16 ± 0.12 s and the gamma band power by 1.15 ± 0.13 s, respectively (*Figure 6A*, see *Figure 6—figure supplement 1*). The peak cross-correlation during contiguous NREM sleep was nearly double that of awake rest: MUA: 0.44 ± 0.06 (GLME, p<1.2 × 10$^{-23}$ compared with awake rest); gamma band: 0.3 ± 0.05 (GLME, p<8 × 10$^{-23}$ compared with awake rest). During NREM, there were similar dynamics in the lag of the hemodynamic response to the MUA as during awake rest (1.28 ± 0.11 s, GLME, p<0.08) and for the gamma band power (1.26 ± 0.12 s, GLME, p<0.15) (*Figure 6B*). The peak cross-correlations during contiguous REM sleep

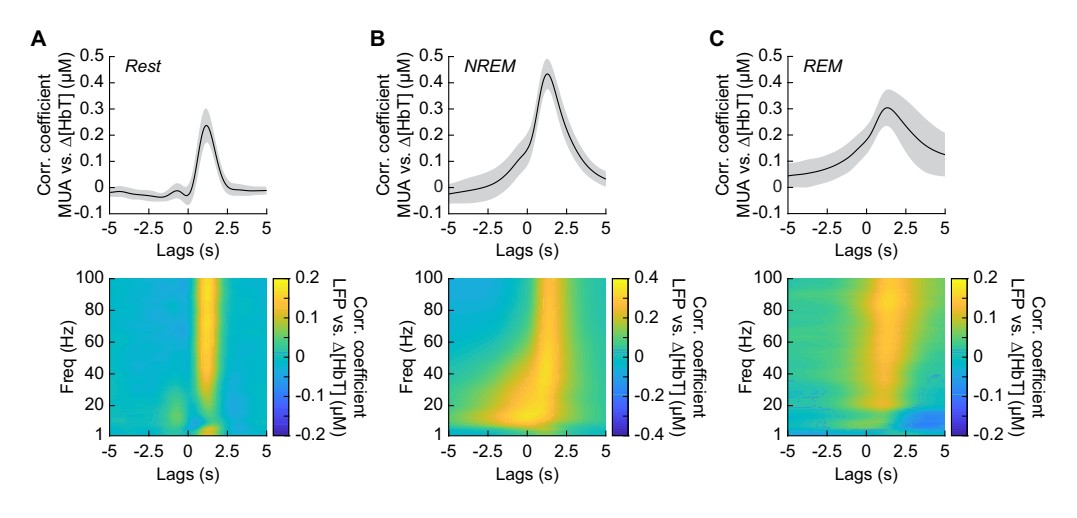

**Figure 6.** Neurovascular coupling is strongest during NREM sleep. Cross-correlation between neural activity and changes in total hemoglobin Δ[HbT] during different arousal states, averaged across hemispheres. MUA power [300–3000 Hz] (top) and the LFP [1–100 Hz] (bottom) were consistently correlated with hemodynamics to varying degrees. (A) Awake rest. (B) Contiguous NREM sleep. (C) Contiguous REM sleep. Shaded regions indicate ±1 standard deviation (n = 14 mice, 28 hemispheres).

The online version of this article includes the following figure supplement(s) for figure 6:

**Figure supplement 1.** Neurovascular coupling dynamics change with arousal state.

were significantly higher than those during awake rest for the MUA (0.32 ± 0.07, GLME, $p < 4.8 \times 10^{-7}$), but not for gamma band power (0.15 ± 0.05; GLME, $p < 0.22$). During REM there was a slight, but significant increase in the lag of the hemodynamic response to the MUA (1.49 ± 0.49 s, GLME, $p < 1.2 \times 10^{-5}$) and for the gamma band power (1.4 ± 0.47 s; GLME, $p < 0.0016$) (*Figure 6C*). However, the distribution of correlations across frequencies differed across states. In addition to a highly correlated gamma band, the correlations during contiguous NREM extended down into the beta band [13–30 Hz]. These results show that the correlation between neural activity and cerebral blood volume change markedly across states, though the hemodynamic lag differences are of order a few hundred milliseconds. We found weak to negative correlations for frequencies below 30 Hz in the awake state and during contiguous REM. In contrast, there were strong positive correlations between lower frequencies of LFP power and the [HbT] in contiguous NREM sleep. This suggests that the positive correlations between lower frequencies of the LFP and vasodilation seen in some studies may be due to the inclusion of NREM sleep in the 'resting-state'.

## Neural and hemodynamic correlations across hemispheres increase during sleep

We then explored how the correlations and coherency of left and right hemisphere [HbT] and gamma band power were affected by arousal state. When quantifying signals in the frequency domain, the lowest resolvable frequency will be the inverse of the event duration. Because there were very few periods of awake rest greater than 10 s in duration (see *Figure 2E*), the lowest frequency for resting periods we could characterize was 0.1 Hz (*Mitra and Pesaran, 1999*). We thus examined all 15 min periods from our data without whisker stimulation that had at least 12 min (>80%) of rfc-Awake arousal state classifications as *alert*, as an extension of the awake resting and awake whisking arousal state into the lower frequencies. We also examined all 15 min periods without whisker stimulation that had at least 12 min (>80%) of rfc-NREM or rfc-REM arousal state classifications as *asleep* to extrapolate the contiguous NREM/REM arousal states into the lower frequencies. Note that the *asleep* condition contains some awake state and transitions, which may explain the lower power in the gamma band envelope above 0.1 Hz than the pure REM and NREM conditions. Lastly, we took every 15 min period without whisker stimulation, regardless of random forest classified arousal state, as *all data*, which has an arousal state distribution similar to *Figure 2B*

and is more representative of the type of data obtained if behavioral monitoring and arousal state classification were not used.

The power spectrum of Δ[HbT], normalized to the peak power in the resting condition, is shown in *Figure 7D* (see *Table 1*, see *Figure 7—figure supplement 1E,F*). For [HbT], there was substantially more power in the lower frequencies for nearly all arousal states. The lower power of the Δ[HbT] power spectra at higher frequencies were consistent with the lowpass nature of the hemodynamic response (*Drew, 2019*; *Silva et al., 2007*; *Winder et al., 2017*; *de Zwart et al., 2005*). We

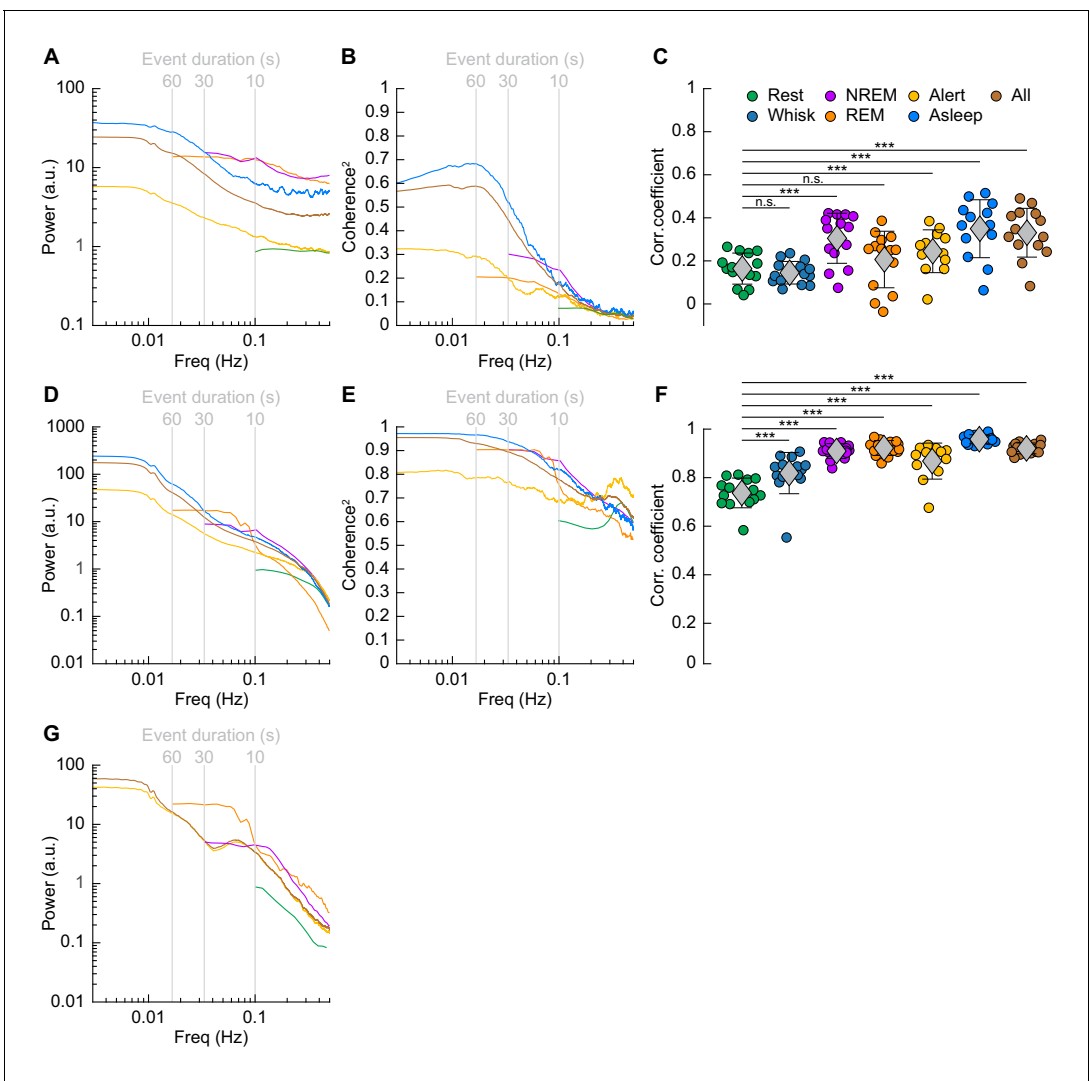

**Figure 7.** Correlations in neural activity and blood volume between hemispheres increase during sleep. (A) Mean gamma band power spectral density during different arousal states. (B) Mean coherence (between hemispheres) in the changes in the envelope (≤1 Hz) of gamma band power [30–100 Hz] between left and right vibrissa cortex during different arousal states. (C) Average gamma band power Pearson's correlation coefficient between left and right vibrissa cortex during different arousal states. Circles represent individual mice and diamonds represent population averages ± 1 standard deviation. (D-F) Same as in A-C except for the changes in total hemoglobin Δ[HbT]. MoC2 between the left and right somatosensory cortex for gamma band power and Δ[HbT] during each arousal state exceeded the 95% confidence level for all frequencies below 1 Hz. (A-F) n = 14 mice (n*two hemispheres in A,D) for all arousal states except Alert: n = 12 mice, Asleep: n = 13 mice. (G) Mean arteriole ΔD/D power spectral density during different arousal states (Rest: n = 6 mice, 29 arterioles, NREM: n = 6 mice, 21 arterioles, REM: n = 5 mice, 10 arterioles, Awake: n = 6 mice, 27 arterioles, All data: n = 6 mice, 29 arterioles). *p<0.05, **p<0.01, ***p<0.001 GLME.

The online version of this article includes the following figure supplement(s) for figure 7:

**Figure supplement 1.** Arousal state dependence of low-frequency neural power and coherence[2].
**Figure supplement 2.** Correlations in neural activity between hemispheres increase during sleep.
**Figure supplement 3.** Arousal state dependence of low-frequency neural power and coherence.

**Table 1.** Spectral power in gamma band, Δ[HbT], and arteriole diameter (ΔD/D) at 0.1 Hz.

| Spectral Power at 0.1 Hz | Awake Rest | Cont. NREM | Cont. REM | Alert | Asleep | All Data |
|---|---|---|---|---|---|---|
| Gamma band Power (a.u.) | 0.9 ± 0.1 | 13.3 ± 37.4 (p<0.01) | 12.8 ± 23.4 (p<0.02) | 1.3 ± 0.8 (p<0.93) | 6.2 ± 8 (p<0.29) | 3.6 ± 2.5 (p<0.58) |
| Δ[HbT] Power (a.u.) | 0.9 ± 0.1 | 6.7 ± 3.4 (p<7.6 × $10^{-19}$) | 4.1 ± 2.6 (p<1.3 × $10^{-7}$) | 2.2 ± 1.3 (p<0.03) | 4.7 ± 2.3 (p<1.3 × $10^{-9}$) | 3.8 ± 1.6 (p<1.3 × $10^{-6}$) |
| ΔD/D Power (a.u.) | 0.9 ± 0.1 | 4.5 ± 3.9 (p<2.7 × $10^{-10}$) | 3.8 ± 3.7 (p<2.1 × $10^{-5}$) | 3.4 ± 2.1 (p<9.3 × $10^{-7}$) | | 3.4 ± 1.9 (p<5.8 × $10^{-7}$) |

Mean ± 1 standard deviation. p-values as a comparison to 'Rest'. Shaded region indicates insufficient data for that arousal state.

examined the power in two frequency bands, 0.1 Hz (the 'vasomotion' frequency *Mateo et al., 2017*), and the ultra-low 0.01 Hz band. The power at 0.1 Hz was highest during contiguous NREM (awake rest: 0.9 ± 0.1 (A.U.); contiguous NREM: 6.7 ± 3.4, GLME vs. awake rest, p<7.6 × $10^{-19}$), and was also higher during contiguous REM than during awake rest (contiguous REM: 4.1 ± 2.6, GLME vs. awake rest, p<1.3 × $10^{-7}$). Extending into the ultra-low frequencies, power in the Δ[HbT] signal at 0.01 Hz was significantly higher in the asleep state than the alert state (176.2 ± 98.9 vs. 36.5 ± 28.6, GLME, p<5.3 × $10^{-10}$). Similar low-frequency dominated power spectra were seen in the diameters of arterioles, with the power at 0.1 Hz during contiguous NREM (4.5 ± 3.9, GLME, p<2.3 × $10^{-8}$) and contiguous REM (5.1 ± 5.7, GLME, p<2.7 × $10^{-7}$) also exceeding those seen during awake rest (0.9 ± 0.1). There was not enough single arteriole data that met the inclusion criteria for the asleep condition for 0.01 Hz (*Figure 7G*, *Table 1*, see *Figure 7—figure supplement 1I,J*).

Previous functional imaging studies have observed marked increases in hemodynamic activity and functional connectivity between brain regions during periods of sleep (*Boly et al., 2012*; *Dang-Vu et al., 2008*; *Fukunaga et al., 2006*; *Horovitz et al., 2008*; *Larson-Prior et al., 2009*; *Mitra et al., 2015*), but how these hemodynamic changes relate to neural activity was unknown.

We then looked at gamma band power, which of all neural signals is most closely related to hemodynamic changes in the awake brain (*Echagarruga et al., 2020*; *Mateo et al., 2017*; *Schölvinck et al., 2010*; *Winder et al., 2017*). The fluctuations in the gamma band power had substantial power at all frequencies (*Figure 7A*, *Table 1*, see *Figure 7—figure supplement 1A,B*), consistent with previous recordings in primates (*Leopold et al., 2003*). Fluctuations in the gamma band power envelope at frequencies near 0.1 Hz have been proposed to drive vasomotion and underly functional connectivity (*Mateo et al., 2017*), and lower frequency fluctuations in the envelope (in the ~0.01 Hz range) could underly changes in connectivity at these frequencies (*Chang and Glover, 2010*). The power in the gamma band envelope at 0.1 Hz was substantially higher during sleep (awake rest: 0.9 ± 0.1; contiguous NREM sleep: 13.3 ± 37.4, GLME, p<0.01; contiguous REM Sleep: 12.8 ± 23.4, GLME, p<0.02). All power normalized by the peak in the power spectrum during rest. There was a similar increase in the gamma band power fluctuations at 0.01 Hz during sleep relative to the awake condition (alert: 5 ± 3.2; asleep: 34 ± 51, GLME, p<0.001). During sleep, the modulations of gamma band power at 0.1 and 0.01 Hz were much larger than during the awake state.

To quantify the relationship of neural activity or [HbT] between the left and right hemispheres, we looked at the magnitude of the coherence squared (MoC2) (*Figure 7B*, *Tables 3* and *4*, see *Figure 7—figure supplement 1C,D*). We used MoC2 because it tells us the amount of variance

**Table 2.** Spectral power in gamma band, Δ[HbT], and arteriole diameter (ΔD/D) at 0.01 Hz.

| Spectral Power at 0.01 Hz | Alert | Asleep | All Data d |
|---|---|---|---|
| Gamma band Power (a.u.) | 5 ± 3.2 | 34 ± 51 (p<0.001) | 21.7 ± 20.6 (p<0.06) |
| Δ[HbT] Power (a.u.) | 36.5 ± 28.6 | 176.2 ± 98.9 (p<5.3 × $10^{-10}$) | 128.9 ± 63.8 (p<7.9 × $10^{-6}$) |
| ΔD/D Power (a.u.) | 34.9 ± 25 | | 45.6 ± 33.6 (p<0.06) |

Mean ± 1 standard deviation. p-values as a comparison to 'Alert'. Shaded region indicates insufficient data for that arousal state.

**Table 3.** Magnitude of Coherence$^2$ of bilateral gamma band and bilateral $\Delta$[HbT] at 0.1 Hz.

| Coherence$^2$ at 0.1 Hz | Awake Rest | Cont. NREM | Cont. REM | Alert | Asleep | All Data |
|---|---|---|---|---|---|---|
| Gamma band (Coherence$^2$) | 0.08 ± 0.06 | 0.26 ± 0.14 (p<4.8 × 10$^{-8}$) | 0.16 ± 0.1 (p<0.006) | 0.13 ± 0.08 (p<0.04) | 0.2 ± 0.11 (p<0.0002) | 0.19 ± 0.09 (p<0.0004) |
| $\Delta$[HbT] (Coherence$^2$) | 0.61 ± 0.11 | 0.86 ± 0.05 (p<1.2 × 10$^{-16}$) | 0.77 ± 0.08 (p<5.7 × 10$^{-10}$) | 0.69 ± 0.14 (p<0.002) | 0.82 ± 0.07 (p<2.9 × 10$^{-13}$) | 0.78 ± 0.06 (p<2.6 × 10$^{-10}$) |

Mean ± 1 standard deviation. p-values as a comparison to 'Rest'.

explained in one signal by the other for any given frequency, making MoC2 equivalent to the R$^2$ between two signals at any given frequency (*Drew et al., 2020*). The MoC2 between left and right hemisphere gamma band power at 0.1 Hz was relatively low. In the 0.01 Hz frequency band, left-right somatosensory cortex gamma band power MoC2 was higher. Interestingly, for all conditions, the MoC2 for the gamma band power was substantially lower than the MoC2 of [HbT] in the same frequency band (*Tables 3* and *4*).

The MoC2 between left and right cortical somatosensory cortex $\Delta$[HbT] was uniformly high (*Figure 7E*, *Tables 3* and *4*, see *Figure 7—figure supplement 1G,H*), though it was higher during contiguous NREM sleep (0.86 ± 0.05, GLME, p<1.2 × 10$^{-16}$) and contiguous REM sleep (0.77 ± 0.08, GLME, p<5.7 × 10$^{-10}$) as compared to awake rest (0.61 ± 0.11). low-frequency $\Delta$[HbT] MoC2 during the asleep state reached near-unity at 0.01 Hz (0.97 ± 0.02, GLME, p<1.5 × 10$^{-5}$), continuously sur-passing those seen during the alert state (0.82 ± 0.15). To compare these results to functional con-nectivity measures, we evaluated the Pearson's correlation coefficient between bilateral $\Delta$[HbT] during each arousal state (*Figure 7F*, *Table 5*). The correlation between the left and right hemi-sphere $\Delta$[HbT] was 0.74 ± 0.06 during the awake resting condition. They were elevated during awake whisking 2–5 s in duration (0.82 ± 0.09, GLME, p<4.8 × 10$^{-8}$), during periods of contiguous NREM (0.9 ± 0.03, GLME, p<9.8 × 10$^{-20}$) and contiguous REM (0.91 ± 0.03, GLME, p<1.4 × 10$^{-21}$). These correlations, however, were not fully explained by the correlations in underlying neural activity, which were much lower. This may be explained by a lower signal-to-noise ratio for neural signals. The cor-relation coefficients for the envelope of bilateral gamma band power [30–100 Hz] (*Figure 7C*, *Table 5*) were relatively small during rest (0.16 ± 0.07), and were not significantly elevated during awake whisking (0.15 ± 0.05, GLME, p<0.47) or during contiguous REM sleep (0.21 ± 0.13, GLME, p<0.11). However, correlations in gamma band power doubled during contiguous NREM sleep (0.3 ± 0.12, GLME, p<6.6 × 10$^{-7}$). Analysis of the cross-hemisphere delta band [1–4 Hz], theta band [4–10 Hz], alpha band [10–13 Hz], and beta band [13–30 Hz] power showed similar trends as the gamma band, with elevated power, MoC2, and Pearson's correlations during contiguous NREM/ REM sleep compared to awake rest (see *Figure 7—figure supplement 2*, *Figure 7—figure supple-ment 3*, *Supplementary file 1*). The exception was the delta band, where the sleeping MoC2 and Pearson's correlations were not statistically different than those during rest. These results show large changes in gamma band power during sleep, but relatively low cross-hemisphere correlations in neu-ral activity. In contrast, the correlation coefficients for the left and right $\Delta$[HbT] were uniformly high (*Table 3*), and significantly elevated by awake whisking. Contiguous NREM and REM sleep substan-tially elevated the $\Delta$[HbT] correlations.

All and all, these result show that sleep enhances the already strong cross-hemisphere coherency and correlation in $\Delta$[HbT]. The increases in $\Delta$[HbT] during sleep are larger than those during whisk-ing/fidgeting behavior, which are the primary drivers of spontaneous hemodynamic signals in the awake mouse (*Drew et al., 2019*; *Drew et al., 2020*; *Winder et al., 2017*). Sleep increases the

**Table 4.** Magnitude of Coherence$^2$ of bilateral gamma band and bilateral $\Delta$[HbT] at 0.01 Hz.

| Coherence$^2$ at 0.01 Hz | Alert | Asleep | All Data |
|---|---|---|---|
| Gamma band (Coherence$^2$) | 0.38 ± 0.27 | 0.69 ± 0.18 (p<7.2 × 10$^{-6}$) | 0.61 ± 0.21 (p<0.0002) |
| $\Delta$[HbT] Power (Coherence$^2$) | 0.82 ± 0.15 | 0.97 ± 0.02 (p<1.5 × 10$^{-5}$) | 0.95 ± 0.03 (p<7.5 × 10$^{-5}$) |

Mean ± 1 standard deviation. p-values as a comparison to 'Alert'.

**Table 5.** Pearson's correlation coefficients of bilateral gamma band and bilateral Δ[HbT].

| Pearson's Correlation Coef. | Awake Rest | Whisking | Cont. NREM | Cont. REM | Alert | Asleep | All Data |
|---|---|---|---|---|---|---|---|
| Gamma band (R) | 0.16 ± 0.07 | 0.15 ± 0.05 (p<0.47) | 0.3 ± 0.12 (p<6.1 × 10⁻⁷) | 0.21 ± 0.13 (p<0.11) | 0.26 ± 0.1 (p<0.0004) | 0.35 ± 0.14 (p<4.7 × 10⁻¹⁰) | 0.33 ± 0.11 (p<8.5 × 10⁻⁹) |
| Δ[HbT] (R) | 0.74 ± 0.06 | 0.82 ± 0.09 (p<3 × 10⁻⁸) | 0.9 ± 0.03 (p<3.6 × 10⁻²⁰) | 0.91 ± 0.03 (p<4.9 × 10⁻²²) | 0.87 ± 0.07 (p<7.9 × 10⁻¹⁵) | 0.96 ± 0.02 (p<8.1 × 10⁻²⁸) | 0.93 ± 0.02 (p<1.1 × 10⁻²⁴) |

Mean ± 1 standard deviation. p-values as a comparison to 'Rest'.

cross-hemisphere correlations and coherency of the LFP envelope as well, but the neural correlations are substantially lower than the hemodynamic correlations.

## Impact of arousal state on total blood volume and neural-vascular coherence

To understand how well we can determine the arousal state (Awake/NREM/REM) from the changes in hemodynamic measurements (Δ[HbT] or ΔD/D), we plotted the probability that the animals were

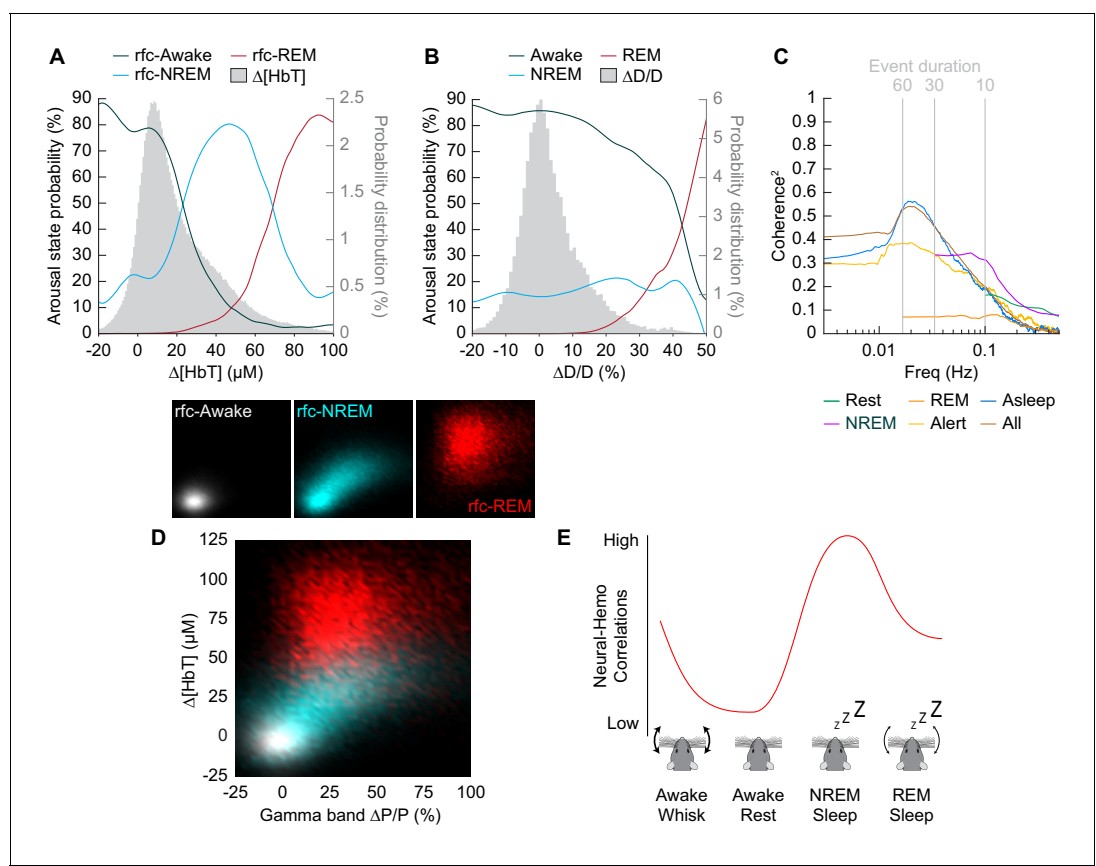

**Figure 8.** Influence of arousal state on vascular correlations to ongoing neural activity. (A) Probability of being in a given arousal state as a function of mean Δ[HbT]. As Δ[HbT] increases, so does the probability that the animal is asleep. Temporal resolution is 5 s, Δ[HbT] bins have a 1 μM resolution. (B) Probability of being in a given arousal state as a function of mean ΔD/D. As ΔD/D increases, the probability that the animal is asleep is not well defined until reaching very large vasodilation. Temporal resolution is 5 s, ΔD/D bins are 1% resolution. (C) MoC2 between the envelope (≤1 Hz) of gamma band power and Δ[HbT] during each arousal state. Three additional states of *alert, asleep,* and *all data* are included to extend into the ultra-low frequencies. n = 14 mice. Alert arousal state n = 12; Asleep arousal state n = 13; All data arousal state n = 14. (D) Relationship between the gamma band power and Δ[HbT] during each arousal state. 5 s resolution. (E) Schematic demonstrating the observed relationship between Δ[HbT] and gamma band power during each arousal state.

The online version of this article includes the following figure supplement(s) for figure 8:

**Figure supplement 1.** Arousal state dependence of low-frequency neural-hemo coherence.

in a given arousal state for a given amount of Δ[HbT] (*Figure 8A*) or ΔD/D (*Figure 8B*). Using only [HbT], the strength of the coupling between spontaneous neural activity and changes in blood volume is relevant to resting-state fMRI. Many studies have looked into this coupling (*Ma et al., 2016*; *Mateo et al., 2017*; *Schölvinck et al., 2010*; *Shmuel and Leopold, 2008*; *Winder et al., 2017*) but the measured strength of this coupling varies from study to study (*Drew et al., 2020*). One possible contribution to the varying strength of the coupling observed between neural and hemodynamic signals could be changes in the arousal state. When solely looking at changes in arteriole diameter (ΔD/D), the variability in arteriole diameter during NREM sleep make it difficult to deduce arousal state. Fluctuations in arteriole diameter during NREM may appear similar to volitional whisking events, which occur every 10–15 s (*Drew et al., 2020*; *Winder et al., 2017*).

The coherence between [HbT] and the gamma band power in various arousal states is shown in *Figure 8C* (other LFP bands are shown in *Figure 8—figure supplement 1*). Coherence at lower frequencies was highest in contiguous NREM sleep and for all data, and lowest during awake rest and during contiguous REM sleep. It may seem puzzling that the coherence is higher when taking all the data together than for any individual subset, but this can be explained by a restriction of range effect when grouping by arousal state. In *Figure 8D*, we show the Δ[HbT] versus the power in the gamma band for all of our 'rfc' data including stimulation. When all the data is considered together, it is clear that there was a robust relationship between neural activity and blood volume. However, as the level of neural activity and [HbT] were different across the arousal states, within any given state this relationship may not be so strong, again due to a restriction of range effect. All in all, we would like to emphasize the observation that correlations between ongoing neural activity and hemodynamic signals appear to be lowest during the awake state and highest during sleep (*Figure 8e*) and is most prominent in the lower frequencies.

## Discussion

Using optical imaging and electrophysiology in head-fixed mice, we found that sleep was associated with large increases in blood volume and arterial dilation in the somatosensory cortex. These vasodilations and increases in cerebral blood volume were very large, up to five times the size of those evoked by sensory stimulation or fidgeting behaviors like whisking. We found that the correlation and coherence between neural activity and blood volume were substantially stronger during NREM sleep than during any other arousal state, as were bilateral correlations and coherency. Because, at least under our imaging conditions, mice frequently and rapidly enter into sleep, the hemodynamic signals due to sleep will dominate over any awake signals due to the very large changes in neural activity and blood volume in the sleeping cortex. The net effect is that, without careful monitoring of arousal state, sleep-related hemodynamics and neural activity will dominate any 'resting-state' study in the un-anesthetized mouse.

We note that care should be taken in interpreting these results. The sleep patterns and depth may be altered by head-fixation. We only imaged cortical blood volume changes, and other areas of the brain may have different patterns of vasodilation (*Braun et al., 1997*; *Townsend et al., 1973*). We used electrophysiology to assay neural activity as it is used in humans (*Buzsáki et al., 2012*), and calcium indicators buffer intracellular calcium, can cause epilepsy, and do not report the majority of spikes even under optimal imaging conditions (*McMahon and Jackson, 2018*; *Steinmetz et al., 2017*; *Theis et al., 2016*). The activity of interneurons that express neuronal nitric oxide synthase (nNOS) can drive vasodilation without causing detectable changes in the LFP (*Echagarruga et al., 2020*; *Krawchuk et al., 2020*; *Lee et al., 2020*), so the overall neural activity captured in the LFP may not reflect the activity of neurons that drive hemodynamic responses. We note there may be a difference in sleep-state identification with local LFP recordings when compared with conventional EEG, which sum electrical signals over a larger area. Previous studies in rats have shown discrete regions of cortex going into periods of 'localized sleep' (*Vyazovskiy et al., 2011*), and in some cases, slow-wave oscillations can be seen during REM sleep in the cortex (*Funk et al., 2016*), either of which may reduce the accuracy of our arousal state classifications. We noted no arousal state discrepancies between left and right somatosensory cortices during manual sleep scoring, however, this does not exclude localized sleep in other cortical regions. The spatial resolution of IOS and our electrodes are of order 100 μm, and any vascular or neural changes on smaller spatial scales would be 'blurred', potentially resulting in lower estimates of neurovascular coupling strength. Additionally,

as hemoglobin is the major absorber of both visible and infrared light in the brain, and changes in the level of hemoglobin can cause decreases in florescence signals (*Haiss et al., 2009*; *Shen et al., 2012*), sleep-related vasodilations attenuate signals from calcium indicators in neurons when visualized with either 1- or 2-photon methods. As there are multiple pathways by which neurons (and astrocytes) communicate with the vasculature (*Attwell et al., 2010*; *Drew, 2019*; *Iadecola, 2017*), it is likely that multiple mechanisms underlly the vasodilation we observed during sleep (*Özbay et al., 2018*).

There are several implications for our work. First, it shows that it is critical to monitor arousal state, particularly in head-fixed mice, as they frequently fall asleep. Arousal state cannot be detected by simple monitoring of the face/eyes, as mice can sleep with their eyes open (*Yüzgeç et al., 2018*), and thus requires additional methods such as a combination of EMG/LFP/whisker tracking. If an animal falls asleep, this will result in larger neurovascular and bilateral correlations than in the awake state. There has been some disagreement as to how well hemodynamic signals track neural activity in the absence of any sensory stimulation (the 'resting-state') (*Drew et al., 2020*). With a few exceptions (e.g. *Winder et al., 2017*), these studies did not carefully monitor arousal state, and it could be that studies finding higher neurovascular correlations have episodes of sleep in them. Similar issues have been noted for human resting-state fMRI studies (*Tagliazucchi and Laufs, 2014*) and with non-human primates (*Cardoso et al., 2019*; *Chang et al., 2016*), so this is likely a ubiquitous problem with any experiment where the subject is unmotivated to stay awake. On a more fundamental level, these results show that the vasodilation seen in the awake brain is smaller than the hemodynamic changes seen during sleep. This vasodilation during sleep may serve to circulate cerebrospinal fluid (*Aldea et al., 2019*; *Fultz et al., 2019*; *Kedarasetti et al., 2020*; *van Veluw et al., 2020*) or some other physiological role.

# Materials and methods

**Key resources table**

| Reagent type (species) or resource | Designation | Source or reference | Identifiers | Additional information |
|---|---|---|---|---|
| Strain, strain background (*M. musculus*) | Strain: C57BL6/J | Jackson Laboratory | Stock No: 000664 | |
| Chemical compound, drug | Fluorescein isothiocyanate–dextran 150 kDa | Sigma-Aldrich | Stock No: FD150S | 100 µL, 5% (weight/volume) |
| Software, algorithm | Data analysis software | MathWorks | MATLAB 2019a | https://github.com/KL-Turner/Turner_Gheres_Proctor_Drew_eLife2020; *Turner et al., 2020* |
| Software, algorithm | IOS and 2PLSM data acquisition software | National Instruments | LabVIEW 2018 | https://github.com/DrewLab/LabVIEW-DAQ; *Drew, 2020* |
| Software, algorithm | 2PLSM data acquisition software | Sutter Instrument | MSCAN 2015 | |
| Other | PFA-coated tungsten microwires | A-M Systems | Stock No: #795500 | |
| Other | PFA-coated 7-strand stainless-steel microwires | A-M Systems | Stock No: #793200 | |

## Animal procedures

This study was performed in accordance with the recommendations in the Guide for the Care and Use of Laboratory Animals of the National Institutes of Health. All procedures were performed in

accordance with protocols approved by the Institutional Animal Care and Use Committee (IACUC) of Pennsylvania State University (protocol # 201042827). All data was acquired from 20 C57BL6/J mice (#000664, Jackson Laboratory, Bar Harbor, ME) comprised of 11 males and nine females between the ages of 3 and 8 months of age. Of these 20 animals, 14 were used in IOS experiments and six were used in two-photon experiments. Any animals that were excluded from a specific analysis are noted. Mice were given food and water ad libitum and housed on a 12 hr light/dark cycle, remaining individually housed after surgery and throughout the duration of experiments. All experiments were performed during the animal's light cycle. Sample sizes are consistent with previous studies (*Drew et al., 2011*; *Huo et al., 2015a*; *Winder et al., 2017*). The experimenters were not blind to the conditions of the experiments, data, or analysis.

## Surgery

### Electrode, EMG, and window implantation procedure for intrinsic optical signal (IOS) imaging experiments

Mice were anesthetized under isoflurane (5% induction, 2% maintenance) vaporized in oxygen during all surgical procedures. The incision site on the scalp was sterilized with betadine and 70% ethanol, followed by the resection of the skin and connective tissue. A custom-machined titanium head bar for head-fixation (https://github.com/DrewLab/Mouse-Head-Fixation) was adhered atop the occipital bone of the skull with cyanoacrylate glue (Vibra-Tite 32402, ND Industries, Clawson, MI) posterior to the lambda cranial suture. A self-tapping 3/32' #000 screw (J.I. Morris, Oxford, MA) was implanted into the center of each frontal bone. The neural recordings were grounded to one of the frontal screws with a stainless-steel wire (#792800, A-M Systems, Sequim, WA). Two ~ 4 mm x ~ 2 mm polished and reinforced thinned-skull windows (*Drew et al., 2010a*; *Shih et al., 2012b*) were bilaterally implanted caudal to the bregma cranial suture above the left and right somatosensory cortices. For each window, the skull was thinned and then sequentially polished with 3F and 4F grit. A PFA-coated tungsten stereotrode (#795500, AM systems, Sequim, WA) was inserted ~700 µm below the pia into the whisker representation of somatosensory cortex (~2 mm caudal,~3–3.5 mm lateral from bregma) at 45 ° from the horizontal along the rostrocaudal axis. A third tungsten stereotrode was implanted ~1500 µm below the pia into the CA1 region of the left hippocampus (~2.5 mm caudal, 4–4.5 mm lateral from bregma) at 45° from the vertical along the mediolateral axis. Each electrode was positioned using a micromanipulator (MP285, Sutter Instruments, Novato, CA) through a small hole made at the edge of the thinned area for the vibrissa electrodes, and slightly caudal the thinned area for the left hemisphere hippocampal electrode. Each hole was sealed with cyanoacrylate glue, and a #0 coverslip (#72198, Electron Microscopy Sciences, Hatfield, PA) was placed atop the thinned portion of the window. The skin above the neck was resected and a pair of PFA-coated 7-strand stainless-steel wires (#793200, AM systems, Sequim, WA) were inserted into each nuchal muscle for EMG recording. The skin was then re-attached to the edge of the occipital bone (Vet-Bond, 3M, St. Paul, MN). Dental cement (Ortho-Jet, Lang Dental, Wheeling, IL) was used to seal the edges of the window and provide structural rigidity to the electrodes, screws, and head bar.

### Electrode, EMG, and window implantation procedure for two-photon laser scanning microscopy (2PLSM) experiments

As described above, mice were anesthetized with isoflurane and a titanium head bar was implanted, along with two frontal screws and ground wire. A third self-tapping 3/32' #000 screw was implanted into the left parietal bone. Instead of bilateral polished and reinforced thinned-skull windows, a single ~4 mm x ~ 5 mm window above the right hemisphere somatosensory cortex is implanted following thinning and polishing. There are no electrodes implanted under the window, as their illumination by the laser causes heating. Tungsten stereotrodes were implanted into the left hemisphere vibrissa cortex and left hemisphere hippocampus in a fashion similar to above. Stainless-steel EMG wires are implanted into the nuchal muscle, and the entire area sealed with dental cement. Following surgery, animals were given 2–3 days to recover before habituation.

## Histology

At the conclusion of the experiments, animals were deeply anesthetized under 5% isoflurane for several minutes and transcardially perfused with heparinized saline, followed by 4% paraformaldehyde.

Fiduciary marks were made at the corner of each cranial window. The extracted brains were put in a solution of 4% PFA/30% sucrose for several days before being coronally section (~90 μm per section) with a freezing microtome. Cytochrome oxidase (CO) staining was performed (c2506, Sigma-Aldrich, St. Louis, MO) to allow visualization of the whisker barrels (*Adams et al., 2018*; *Drew and Feldman, 2009*) and the hippocampal layers (*Figure 1—figure supplement 1D,E*). All histological schematics (*Figure 1C*, *Figure 3C*) as well as histological overlays (*Figure 1—figure supplement 1D,E*) were adapted from the mouse brain in stereotactic coordinates, 3rd Edition (*Franklin and Paxinos, 2007*).

## Physiological data acquisition

All IOS and 2PLSM experiments were performed in sound-dampening boxes. IOS data were acquired with a custom LabVIEW program (v18.0, National Instruments, Austin, TX). 2PLSM data were acquired with Sutter MCS software (Sutter Instruments, Novato, CA) and a custom LabVIEW program designed to synchronize with the Sutter MCS software. Both custom LabVIEW programs are available at https://github.com/DrewLab/LabVIEW-DAQ.

### Habituation

Mice were gradually acclimated to being head-fixed over the course of three habituation sessions of increasing duration. During the initial habituation session (15–30 min in duration), animals were not exposed to any whisker stimulation and the efficacy of the cortical, hippocampal, and EMG electrodes was determined. If the electrical recordings were patent and the mouse tolerated head-fixation, animals were habituated for two more sessions of 60 and 120 min. During these subsequent sessions, the whiskers were stimulated with directed air puffs. Following habituation, IOS animals were run for six imaging sessions lasting of 3–5 hr, and 2PLSM animals were run for up to six imaging sessions depending on the quality of the thinned-skull window.

### Intrinsic optical signal (IOS) imaging

Mice were briefly (<1 min) anesthetized with 5% isoflurane and transferred to the head-fixation apparatus with the body being supported by a clear plastic tube. Animals were given 30 min to wake up prior to data collection to allow recovery from isoflurane (*Shirey et al., 2015*). Changes in total blood volume were measured by illuminating each cranial window with two collimated and filtered 530 ±5 nm LEDs (FB530-10 and M530L3, Thorlabs, Newton, NJ). The 530 nm wavelength is an isosbestic point in which oxy- and deoxy- hemoglobin absorb the light equally. We use the changes in the amount of light reflected from the surface of the brain as a measurement of total hemoglobin concentration. The reflected light is imaged with a Dalsa 1M60 Pantera CCD camera (Phase One, Cambridge, MA) positioned above the mouse's head. The magnification of the lens (VZM 300i, Edmund Optics, Barrington, NJ) allows simultaneous collection of data from both the left and right cranial windows. The light entering the camera (green) was filtered using a mounted filter (#46540, Edmund Optics, Barrington, NJ) to remove the red light used in whisker tracking. Images for tracking changes in total hemoglobin (256 × 256 pixels, 15 μm per pixel, 12-bit resolution) were acquired at 30 frames/second (*Huo et al., 2015a*; *Winder et al., 2017*).

### Two-photon laser scanning microscopy (2PLSM)

Mice were briefly (<1 min) anesthetized with 5% isoflurane and retro-orbitally injected with 100 μL of 5% (weight/volume) fluorescein isothiocyanate–dextran (FITC) (FD150S, Sigma-Aldrich, St. Louis, MO) dissolved in sterile saline. Mice were then head-fixed in a similar set-up as during IOS experiments and given 30 min to wake up prior to data collection (*Shirey et al., 2015*). Imaging was done on a Sutter Movable Objective Microscope with a CFI75 LWD 16X W objective (Nikon, Melville, NY) and a MaiTai HP Ti:Sapphire laser ( Spectra-Physics, Santa Clara, CA) tuned to 800 nm. Individual pial (n = 25) and penetrating (n = 4) arterioles were imaged (five frames/second) in 16 min intervals with a power of 10–20 mW (measured exiting the objective). All arterioles measured were in somatosensory cortex and in or near the whisker vibrissa representation.

### Electrophysiology

Neural activity was recorded simultaneously in both IOS and 2PLSM as the differential potentials between the two leads of either the PFA-coated tungsten microwires (#795500, A-M Systems,

Sequim, WA) (*Huo et al., 2014*; *Winder et al., 2017*) for cortical and hippocampal stereotrodes. EMG activity was identically recorded with PFA-coated 7-strand stainless-steel microwires (#793200, A-M systems, Sequim, WA). Stereotrode tungsten microwires were threaded through polyimide tubing (#822200, A-M Systems, Sequim, WA) giving an interelectrode spacing of ~100 μm. The tungsten microwires were crimped to gold pin connectors, with impedances typically between 70 and 120 kΩ at 1 kHz. EMG stainless-steel microwires were fabricated in a similar fashion, but with an interelectrode spacing of several mm. Each signal was amplified and hardware bandpass filtered between 0.1 Hz and 10 kHz (DAM80, World Precision Instruments, Sarasota, FL) and then digitized at 20 kHz (PCIe-6341 for IOS experiments, PCIe-6321 and PCIe-6353 for 2PLSM experiments, National Instruments, Austin, TX).

### Laser Doppler flowmetry (LDF)

In a subset of IOS animals (n = 8, one animal excluded) a laser Doppler probe (OxyLab LDF, Oxford Optronix, Abingdon, United Kingdom) was aligned above the right hemisphere barrel cortex to record changes in bulk flow and was digitized at 20 kHz.

### Whisker stimulation

Mice were stimulated with brief (0.1 s), randomized puffs of air (10 PSI via an air regulator, Wilkerson R03-02-000, Grainger, Lake Forest, IL) to either the left vibrissae, right vibrissae, or an auditory control (with 1:1:1 ratio) every 30 s for the first ~60 min of imaging. The stimuli were directed to the distal ends of the whiskers, parallel to the face so as to avoid stimulating other parts of the body/face. Each stimulus was controlled with a solenoid actuator valve (2V025 ¼, Sizto Tech Corporation, Palo Alto, CA). Only mice undergoing IOS imaging underwent whisker stimulation.

### Behavioral measurements

In both IOS and 2PLSM experiments, the right vibrissae were diffusely illuminated from below by either a 625 nm light (#66–833, Edmund Optics, Barrington, NJ) during IOS experiments, or with a 780 nm LED (M780L3, Thorlabs, Newton, NJ) during 2PLSM experiments. In both experimental setups, a Basler ace acA640-120gm camera (Edmund Optics, Barrington, NJ) with a 18 mm DG Series FFL lens (#54–857, Edmund Optics, Barrington, NJ) acquired images of the whiskers (30 × 350 pixels) at a nominal rate of 150 frames/second. The image was narrow enough to only show the whiskers as dark lines on a bright background, with the average whisker angle being estimated using the Radon transform (https://github.com/DrewLab/Whisker-Tracking; *Drew et al., 2010b*). In addition to whisker tracking, animal motion inside the tube was measured using a pressure sensor (Flexiforce A201, Tekscan, Boston, MA) was amplified (Model 440, Brownlee Precision (NeuroPhase), Santa Clara, CA for IOS experiments, Model SR560, Stanford Research Systems, Sunnyvale, CA for 2PLSM experiments) and digitized at 20 kHz by the same acquisition device(s) previously described for the electrophysiology data. For both whisker acceleration and pressure sensor data, a threshold was manually set to establish when the animal behaved. A webcam (LifeCam Cinema, Microsoft, Redmond WA for IOS experiments, ELP 2.8 mm wide angle IR LED Infrared USB camera for 2PLSM experiments) was used to monitor the animal's behavior during data acquisition via a real-time video stream in the LabVIEW data acquisition program. A Basler ace acA640-120gm camera (Edmund Optics, Barrington, NJ) with a 75 mm DG Series FFL Lens (#54–691, Edmund Optics, Barrington, NJ) acquired an image of the eye (200 × 200 pixels) at a nominal rate of 30 frames/second during IOS experiments. The eye was illuminated with a 780 nm LED (M780L3, Thorlabs, Newton, NJ).

## Data analysis

Data analysis was conducted with code written by KLT, KWG, and PJD (MathWorks, MATLAB 2019b, Natick, MA).

## Alignment of region of interest (ROI) over whisker vibrissa cortex in IOS data

To focus on blood volume changes in the whisker representation of somatosensory cortex, a 1 mm diameter circle was manually placed over the thinned-skull window's region of pixels that were most correlated to that hemisphere's gamma band power during the first 15–60 min of data of each

imaging session (MATLAB function(s): butter, zp2sos, filtfilt, xcorr) (*Figure 1—figure supplement 1A–C*). For each hemisphere (n = 28) this region was typically located in the most caudal, lateral corner of the window consistent with the anatomical location of vibrissa cortex and implantation site of the stereotrode, which remained consistent across all days of imaging. The location of the circular ROI and electrode was verified histologically through the alignment of the electrode path and fiduciary marks with respect to the layer IV CO stain. The reflectance in the circular ROI of pixels was averaged together. To correct a slow drift in the CCD camera's sensitivity to light over several hours, a two-exponent function (MATLAB function(s): fit) was fit to the slow drift of a region of interest over the cement. The profile of this exponential function was then used to remove the slow exponential drift of the mean pixel reflectance over time (see *Figure 1—figure supplement 9*).

## Two-photon laser scanning microscopy imaging processing

Individual stack frames from 2PLSM were corrected for x-y motion artifacts and aligned with a rigid registration algorithm (*Drew et al., 2011*; *Gao et al., 2015*). Imaging periods with excessive z-plane motion artifacts were excluded from analysis. A rectangular box was manually drawn around a straight, evenly-illuminated segment of the vessel and the pixel intensity was averaged along the long axis and used to calculate the vessel's diameter from the full-width at half-maximum (https://github.com/DrewLab/Surface-Vessel-FWHM-Diameter; *Drew et al., 2011*). The diameter of penetrating arterioles was calculated using the thresholding in Radon space (TiRS) algorithm (https://github.com/DrewLab/Thresholding_in_Radon_Space; *Gao and Drew, 2014*; *Gao et al., 2015*) .

## Whisker motion quantification

Images of the mouse's vibrissae were converted into a relative position (angle) by applying the Radon transform (MATLAB function(s): radon). The peaks of the sinogram corresponded to the position and the angle of the whiskers in the image. The average whisker angle was extracted as the angle of the sinogram with the largest variance in the position dimension (https://github.com/DrewLab/Whisker-Tracking; *Drew et al., 2010b*). Vibrissae angles in dropped camera frames were filled with linear interpolation between the nearest valid points (MATLAB function(s): interp1). Whisker angle was lowpass filtered (<20 Hz) using a second-order Butterworth filter and then resampled down to 30 Hz (MATLAB function(s): butter, zp2sos, filtfilt, resample). To identify periods of whisking, whisker acceleration was obtained from the second derivative of the position and binarized with an empirically chosen acceleration threshold for a whisking event. Acceleration events that occurred within 0.1 s of each other were linked and considered as a single whisking bout.

## Movement quantification

Movement data from the pressure sensor was digitally lowpass filtered (<20 Hz) using a second-order Butterworth filter and then resampled down to 30 Hz (MATLAB function(s): butter, zp2sos, filtfilt, resample). To identify movement events, the force sensor data was binarized in a similar fashion to that of the whisker acceleration by setting an empirically defined threshold.

## Heart rate detection

During IOS experiments, the heart rate was detected through the time-frequency spectrogram (3.33 s window, 1 s step size, [2,3] tapers) of the hemodynamic signal (Chronux toolbox, version 2.12 v03). The heart rate was identified as the frequency with the maximum spectral power in the 5–15 Hz band. This signal was then averaged between the two hemispheres, and digitally lowpass filtered (<2 Hz) using a third-order Butterworth filter (MATLAB function(s): butter, filtfilt).

## Neural data and spectrograms

Neural signals (cortical, hippocampal) were subdivided into six frequency bands as follows: delta [1–4 Hz], theta [4–10 Hz], alpha [10–13 Hz], beta [13–30 Hz], gamma [30–100 Hz], and multi-unit activity (MUA) [300–3000 Hz]. Each neural signal was digitally bandpass filtered from the raw data using a third-order Butterworth filter. The data was then squared and lowpass filtered (<10 Hz) using a third-order Butterworth filter, and resampled down to 30 Hz (MATLAB function(s): butter, zp2sos, filtfilt, resample). Several sets of time-frequency spectrograms with varying characteristics were calculated for each neural signal to be utilized in different analysis (Chronux toolbox, version 2.12 v03, function:

mtspecgramc). A 5 s window with 1/5 s step size and [5,9] tapers, a 1 s window with 1/10 s step size and [1,1] tapers, and a 1 s window with 1/30 s step size and [5,9] tapers. All time-frequency spectrograms had the same passband of 1 to 100 Hz to encompass the local field potential (LFP).

## Electromyography (EMG)

Electrical activity from the nuchal (neck) muscles was digitally bandpass filtered (300 Hz – 3 kHz) using a third-order Butterworth filter. The signal was then squared and convolved with a Gaussian kernel with a 0.5 s standard deviation, log transformed, and resampled down to 30 Hz (MATLAB function(s) butter, zp2sos, filtfilt, gausswin, log10, conv, resample).

## Laser Doppler flow velocimetry (LDF)

Microvascular perfusion data was resampled down to 30 Hz and digitally lowpass filtered (<1 Hz) using a fourth-order Butterworth filter (MATLAB function(s) butter, zp2sos, filtfilt, resample).

## Establishment of awake rest and baseline

In order to establish a resting baseline and exclude any drowsy or sleeping data, long periods of *true awake* (typically >1 min) were manually identified during each day's imaging session. Resting events of 5 s or greater were taken from these *true awake* periods and were defined by an absence of whisker stimulation, whisker movement, or detectable body movement for both IOS and 2-photon experiments. Only data from these pre-screened periods of clear wakefulness were used in subsequent baseline calculation as well as in the awake rest and awake whisking arousal state comparisons in subsequent analysis. Periods of rest from *true awake* were thus identified from each imaging session and averaged across time, giving a single baseline value per day for the hemodynamic reflectance signal (IOS or 2P), neural signals, EMG, LFD (if present), and neural spectrograms. For IOS experiments, the baseline reflectance of each day was used to convert changes in reflectance into changes in total hemoglobin (Δ[HbT]) using the Beer-Lambert law (*Ma et al., 2016*).

## Sleep scoring methodology

The data was divided into 5 s bins and classified as rfc-Awake, rfc-NREM, or rfc-REM using a random forest classification model. The model consisted of a 'bagging' (bootstrap aggregation) of 128 decision trees where each tree is grown with an independent bootstrapped replica of the input data (MATLAB function(s): TreeBagger). 128 trees were chosen as sufficient where the out-of-bag-error asymptotes as a function of the number of total trees. The model used the largest mean cortical LFP power from the two hemispheres in the delta band power [1–4 Hz], beta band power [13–30 Hz], and gamma band power [30–99 Hz]. The mean hippocampal theta band power [4–10 Hz], the mean normalized EMG power, the mean heart rate, and the total number of binarized whisking events were also included. To train the random forest classification models, all of the data from the first (session 1) and last (session 6) was manually scored 5 s at a time as either rfc-Awake, rfc-NREM, or rfc-REM based on the known behavioral and electrophysiology characteristics of the various sleep states. For example, an increase in cortical delta band power with a low heart rate and little whisker motion is associated with NREM sleep, while an increase in hippocampal theta band power with a low EMG muscle tone is associated with REM sleep. Half of the manually scored data (1/6 of the total amount) was used to train the random forest classification model, with the other half being used to further validate the model's accuracy (see *Figure 2—figure supplement 2*). The rfc-Awake class from the random forest classification model will include all awake resting data, whisking behavior, as well as all of the 'drowsy' data where the animals was transitioning between states that did not clearly fall into the NREM or REM categories. For these reasons, quantifications of awake rest, awake whisking, and awake stimulation were taken from manually verified periods of wakefulness (a subset of rfc-Awake) to reduce contamination.

For data to be classified as either a contiguous NREM or REM sleep epoch, it requires 6 consecutive 5 s bins (30 s) for contiguous NREM or 12 consecutive 5 s bins (60 s) for contiguous REM. This filter ensures that only very clear NREM and REM sleep events make it into the final data sets that are used in the relevant analysis. It also provides a minimum length for each event for analysis that require for all data to be the same length (such as cross-correlations, coherence, and power spectra). To prevent REM events of several minutes in duration from occasionally being broken up into

multiple separate events, up to 10 s of data in-between rfc-REM classifications were linked after model scoring. While the majority of contiguous NREM/REM epochs occurred in the absence of whisker stimulation, the epochs that did occur in the presence of whisker stimulation were excluded from these contiguous sleep categories. As the total duration of the 2-photon data was substantially less than that of IOS experiments, it was all manually scored. In our experience, the mice tended to sleep less during 2-photon imaging than during IOS, likely because of the high-frequency noise from the scan mirrors and the lack of illumination (mice are nocturnal) (*Peirson et al., 2018*; *Pilorz et al., 2016*).

### Sleep model accuracy validation

The out-of-bag error during random forest classification model training provides an initial estimate on the model's classification accuracy, where *out-of-bag* refers to the mean classification error using training data from the trees that do not contain the data in their bootstrap sample (MATLAB function(s): oobError). The out-of-bag error of each model's training data is then compared to the mean out-of-bag error from 100 randomly shuffled training data sets, which is analogous to random chance. A table of each animal's out-of-bag error and randomly shuffled out-of-bag error can be found in *Supplementary file 1*. The mean out-of-bag error across all animal models was 7.1 ± 1.4% and the mean error across the 100 randomly shuffled data sets across all animals was 36.1 ± 10.6%. Each model was then evaluated on a second, unseen data set composed of the alternating 15 min periods that were manually scored but not used in model training. The model's scores were then compared to the manual scores combined across all IOS animals are summarized in a confusion matrix (see *Figure 2—figure supplement 2*) (MATLAB function(s): confusionchart). Across all animals, the overall accuracy was 91.3%.

### Distribution of arousal state classifications

A hypnogram for each IOS animal (representative animal, *Figure 2A*) was generated to visualize the frequency and duration of sleeping events throughout each imaging session, as well as pick up any on discrepancies between manual and model scoring. The percentage each animal spent in each of the three model's classification states ('rfc-Awake, rfc-NREM, rfc-REM) was averaged across IOS animals (*Figure 2B*) and is shown ratiometrically for individual animals in a ternary plot (*Figure 2C*). The probability distribution of heart rate (*Figure 2F*), whisker variance (*Figure 2G*), and EMG power (*Figure 2H*) during each classification was evaluated by taking the mean (or variance for whisker angle) of each 5 s bin and combining the data from all animals, with no within animal averaging. rfc-NREM and rfc-REM were compared to the rfc-Awake classification to evaluate statistical significance (*Figure 2—figure supplement 1*) by averaging the data (heart rate, whisker variance, EMG) first within animals, and then across animals. Error bars show the standard deviation (n = 14 mice).

### Determination of awake probability

The probability of a mouse being in a given arousal state as a function of imaging time (*Figure 2D*) was evaluated by concatenating all 5 s sleep scores. The probability of an animal being in a given arousal state was then averaged across each IOS animal's data set (consisting of 6 imaging sessions of bilateral imaging), and fit with a single exponential fit (MATLAB function(s): fit) for rfc-Awake, rfc-NREM, and rfc-REM. These three exponentials were then averaged across all 14 animals. Because the recording did not start until 30 min after the start of head-fixation, the animal's probability of being awake at 'time 0' was not 100%. The probability of an animal being asleep as a function of the duration of quiescence (*Figure 2E*) was done by binning the individual event to the appropriate 5 s bin (a 7.5 s quiescent event falls in the 5–10 s bin). The approximate time and duration of each individual event was shifted in time to the corresponding sleep scoring bin(s). If any of the bins contain a score of rfc-NREM or rfc-REM, then the individual event is considered asleep. The number of rfc-Awake events in each time increment is divided by the total number of events for each animal, fit with a single exponential (MATLAB function(s): fit), and averaged across animals.

### Mean heart rate during different states

The mean heart rate during each arousal state was taken during awake resting events (≥10 s), moderate awake whisking events (2–5 s in duration), contiguous NREM sleep events (≥30 s), and

contiguous REM sleep events (≥60 s). All awake resting periods and awake whisking events were taken from data at least 5 s after a whisker stimulus, with the mean value of the whisking heart rate taken between the initiation of the whisk (time 0) through 5 s. All awake resting events and awake whisking events occurred within the manually defined *true awake* periods outlined previously in the *Establishment of awake rest and baseline* section to exclude drowsy behavior. All arousal state events were averaged within their individual time series. All arousal state events were then averaged within animals before being averaged across animals (*Figure 2I*). Error bars show the standard deviation.

## Arousal state transitions

Transitions from rfc-Awake to rfc-NREM, rfc-Awake to rfc-NREM, rfc-NREM to rfc-REM, and rfc-REM to rfc-Awake (*Figure 4A–D*) were taken by averaging all the events within an animal that had six consecutive 'rfc' arousal state scores (30 s) of one arousal state of interest followed by six consecutive scores of the other. Hemodynamic (Δ[HbT]), normalized EMG, normalized cortical LFP, and normalized hippocampal LFP data from each arousal state transition was extracted at the corresponding time indices and averaged together within animals. Bilateral hemodynamic data and bilateral cortical LFP data from the cortical hemispheres was averaged together into one value (n = 14 mice, 28 hemispheres). Cortical and hippocampal LFP spectrograms used were those with parameters of 1 s window, 1/10 s step size, and [5,9] tapers. Hemodynamic ([HbT]) data was digitally lowpass (<1 Hz) filtered using a fourth-order Butterworth filter (MATLAB function(s): butter, zp2sos, filtfilt). Transitions from each animal were averaged together, error bars for the hemodynamic and EMG show standard deviation. Due to the limited amount of REM sleep data from 2-photon mice, NREM to REM (*Figure 4E*) and REM to Awake (*Figure 4F*) for 2-photon data (n = 5 mice, eight arterioles) was taken as the 30 s prior and 30 s post each valid contiguous REM event. Arteriole transitions were smoothed with a 10th-order one-dimensional median filter (MATLAB function(s): medfilt1).

## Mean Δ[HbT] during different arousal states

The mean change in total hemoglobin in each cortical hemisphere (*Figure 5A*) during each arousal state was taken for awake resting events (≥10 s), awake whisking events (2–5 s in duration), awake whisker stimulation, contiguous NREM sleep events (≥30 s), and contiguous REM sleep events (≥60 s). Awake resting and awake whisking events were required to be at least 5 s after a whisker stimulus, with the mean value of the whisking behavior taken between the initiation of the whisk (time 0) through 5 s and contralateral stimuli taken as the mean for 1–2 s after the stimulus. Awake resting, awake whisking, and awake whisker stimuli events occurred within the manually defined *true awake* periods outlined previously in the *Establishment of awake rest and baseline* section to exclude drowsy periods. Each arousal state event was digitally lowpass filtered (<1 Hz) with a fourth-order Butterworth filter (MATLAB function(s): butter, zp2sos, filtfilt) and then averaged within their individual time series. The mean of the 2 s of data prior to the onset of whisking/stimulation were subtracted from whisking events. All arousal state events were then averaged within animals before being averaged across animals. Error bars show the standard deviation. The histograms showing the probability distribution of mean change in total hemoglobin (*Figure 5D*) is for all data (30 Hz resolution) from each individual arousal state event from all animals, with no averaging between arousal states within or across animals.

## Mean arteriole diameter during different arousal states

The mean vessel diameter in each arteriole (*Figure 5B*) during each arousal state was taken during awake rest events (≥10 s), awake whisking events (2–5 s in duration), contiguous NREM sleep events (≥30 s), and contiguous REM sleep events (≥60 s). Awake resting and awake whisking events occurred at least 5 s after a whisker stimulus, with the mean value of the whisking diameter taken between the initiation of the whisk (time 0) through 5 s. Awake resting and awake whisking events occurred within the manually defined *true awake* periods outlined previously in the *Establishment of awake rest and baseline* section to exclude drowsy behavior. All arousal state events were digitally lowpass filtered (<1 Hz) with a fourth-order Butterworth filter (MATLAB function(s): butter, zp2sos, filtfilt) and then the mean value was taken within their individual time series. All arousal state events were then averaged within individual arterioles before being averaged across all arterioles from all

animals combined. Error bars show the standard deviation. The histograms showing the probability distribution of arteriole diameter (*Figure 5E*) is taken from all arterioles (5 Hz resolution) with no averaging between arousal states within or across arterioles.

## Mean laser Doppler flow velocimetry during different arousal states

The mean LDF (*Figure 5C*) during each arousal state was taken during awake resting events (≥10 s), awake whisking events (2–5 s in duration), contiguous NREM sleep events (≥30 s), and contiguous REM sleep events (≥60 s). Awake resting and awake whisking events occurred at least 5 s after a whisker stimulus, with the mean value of the whisking flow taken between the initiation of the whisk (time 0) through 5 s. Awake resting and awake whisking events occurred within the manually defined *true awake* periods outlined previously in the *Establishment of awake rest and baseline* section to exclude drowsy behavior. All arousal state events were digitally lowpass filtered (<1 Hz) with a fourth-order Butterworth filter (MATLAB function(s): butter, zp2sos, filtfilt) and then averaged within their individual time series. All arousal state events were then averaged within animals before being averaged across animals. Error bars show the standard deviation (n = 8 mice, one mouse excluded due to poor signal). The histograms showing the probability distribution of mean change in flow (*Figure 5F*) is for all data (30 Hz resolution) from each individual arousal state event from all animals, with no averaging between arousal states within or across animals.

## Cross-correlations during different arousal states

The cross-correlation between multi-unit activity (MUA) and change in total hemoglobin Δ[HbT] in each cortical hemisphere during each arousal state was taken during awake resting events (≥10 s), contiguous NREM sleep events (≥30 s), and contiguous REM sleep events (≥60 s). Awake resting events occurred at least 5 s after a whisker stimulus and occurred within the manually defined *true awake* periods outlined previously in the *Establishment of awake rest and baseline* section to exclude drowsy behavior. All arousal state events' MUA and Δ[HbT] data were mean-subtracted and digitally lowpass filtered (<1 Hz) with a fourth-order Butterworth filter (MATLAB function(s): butter, zp2sos, filtfilt) and then truncated to the minimum arousal state length so that all events were the same length. Cross-correlation analysis was run for each arousal state (MATLAB function(s): xcorr) with a ± 5 s lag time and averaged across arousal state events within each animal hemisphere and then across all animal hemispheres (n = 14 mice, 28 hemispheres). The cross-correlation between LFP and Δ[HbT] was taken as the cross-correlation between an Δ[HbT] event and each frequency band of the cortical spectrogram with parameters of 1 s window, 1/30 s step size, and [1,1] tapers (*Figure 6*). The resulting cross-correlation matrices (lag time x frequency) were then averaged across all arousal state events within each hemisphere and then across all hemispheres.

## Power spectra and coherence of neural and hemodynamic signals

The coherence between left and right hemisphere [HbT], envelopes of each LFP band (delta, theta, alpha, beta, gamma), or within-hemisphere [HbT] and the power in an LFP band during each state was taken during awake resting events (≥10 s), contiguous NREM sleep events (≥30 s), contiguous REM sleep events (≥60 s), alert data (15 min), asleep data (15 min), and all data (15 min). Awake resting events occurred at least 5 s after a whisker stimulus and occurred within the manually defined *true awake* periods outlined previously in the *Establishment of awake rest and baseline* section. All arousal state events were mean-subtracted and digitally lowpass filtered (<1 Hz) with a fourth-order Butterworth filter (MATLAB function(s): butter, zp2sos, filtfilt) and then truncated to the minimum arousal state length. Coherence analysis was run for each data type during each arousal state (tapers [3,5], pad = 1, Chronux toolbox, version 2.12 v03, function: coherencyc) and averaged across animals. Error bars show the standard deviation. Power spectrum analysis was run for each data type as well as arteriole diameter during each arousal state in each cortical hemisphere (tapers [3,5], pad = 1, Chronux toolbox, version 2.12 v03, function: mtspectrumc) and averaged across animals. For *Figure 7*, neural power spectra were normalized by the peak of the power spectrum in the resting state for each hemisphere before averaging across hemispheres. This normalization accounts for any variation in impedance across electrodes. Data was padded to the second next highest power of 2. Error bars show the standard deviation (n = 14 mice, 28 hemispheres for IOS, n = 6 mice for 2PLSM).

## Pearson's correlation coefficients during different arousal states

The Pearson's correlation coefficient between bilateral cortical changes in total hemoglobin, bilateral envelopes of each discrete LFP band (delta, theta, alpha, beta, gamma) during each arousal state was taken during awake resting events (≥10 s), awake whisking events (2–5 s in duration), contiguous NREM sleep events (≥30 s), contiguous REM sleep events (≥60 s), alert data (15 min), asleep data (15 min), and all data (15 min). Awake resting and whisking events occurred at least 5 s after a whisker stimulus, with the correlation value of the whisking behavior taken between the initiation of the whisk to 5 s later. All resting events and whisking events occurred within the manually defined *true awake* periods outlined previously in the *Establishment of awake rest and baseline* section. All arousal state events for each bilateral data type were mean-subtracted and digitally lowpass filtered (<1 Hz) with a fourth-order Butterworth filter (MATLAB function(s): butter, zp2sos, filtfilt) and then take the Pearson's correlation coefficient (MATLAB function(s): corrcoef) within each time series. All correlation coefficients for each bilateral data type during each arousal state were then averaged within animals and then averaged across animals. Error bars show the standard deviation.

## Probability of sleep as a function of hemodynamics

The probability of an arousal state classification with respect to [HbT] (*Figure 8A*) or arteriole diameter (ΔD/D) (*Figure 8B*) was determined by taking the mean of each 5 s epoch and binning the value into a histogram (1 μM Δ[HbT] or 1% ΔD/D). Each of the three arousal state classifications (rfc-Awake, rfc-NREM, rfc-REM for IOS, manual versions for 2PLSM) corresponding to each bin was summed over the total number of counts per bin to create a probability curve. Each curve was then smoothed with a 10-point median filter (MATLAB function(s): medfilt1). All 5 s bins were weighted equally with no grouping or averaging within animals (n = 14 mice, 28 hemispheres for IOS, n = 6 mice for 2PLSM).

## [HbT]-Gamma relationship

The mean gamma band power (ΔP/P) and Δ[HbT] in each cortical hemisphere during each arousal state classification (rfc-Awake, rfc-NREM, rfc-REM) was taken from all IOS animals. A 2D histogram (MATLAB function(s): histogram2) was made comparing the mean gamma band power (ΔP/P) vs. the mean Δ[HbT] of each individual classification (*Figure 8D*) to highlight the clusters of each arousal state class.

## Statistical analysis

All statistical comparisons were evaluated using generalized linear mixed-effects models (MATLAB function(s): fitglme). The arousal state was treated as a fixed effect, the mouse identity as a random effect, and mouse identity:hemisphere/arteriole combination treated as an interaction. For example, the formula for *Figure 5A*: 'HbT ~1 + ArousalState + (1|Mouse) + (1|Mouse:Hemisphere)' evaluates the mean changes in [HbT] (n = 14 mice, 28 hemispheres) of each arousal state (fixed effects) with the mouse (random effect) and an interaction between the mouse and hemisphere (left vs. right hemispheres are not fully independent). Each arousal state was compared to either the 'rfc-Awake', 'awake rest', or 'alert' condition as the intercept term, depending on the analysis.

## Acknowledgements

We thank X Liu, A Shih, A Winder, and N Zhang for comments and discussion on the manuscript and F Bahari for advice on electromyography. This work was supported by NIH grants R01NS078168 and R01NS079737 to PJD.

## Additional information

### Funding

| Funder | Grant reference number | Author |
| --- | --- | --- |
| National Institutes of Health | R01NS078168 | Patrick J Drew |
| National Institutes of Health | R01NS079737 | Patrick J Drew |

The funders had no role in study design, data collection and interpretation, or the decision to submit the work for publication.

### Author contributions
Kevin L Turner, Conceptualization, Resources, Data curation, Software, Formal analysis, Validation, Investigation, Visualization, Methodology, Writing - original draft; Kyle W Gheres, Resources, Software, Methodology, Writing - review and editing; Elizabeth A Proctor, Formal analysis, Supervision, Writing - review and editing; Patrick J Drew, Conceptualization, Data curation, Formal analysis, Supervision, Funding acquisition, Visualization, Writing - original draft, Project administration, Writing - review and editing

### Author ORCIDs
Kevin L Turner (iD) https://orcid.org/0000-0002-3044-7079
Kyle W Gheres (iD) https://orcid.org/0000-0001-7568-9023
Elizabeth A Proctor (iD) https://orcid.org/0000-0002-7627-2198
Patrick J Drew (iD) https://orcid.org/0000-0002-7483-7378

### Ethics
Animal experimentation: This study was performed in accordance with the recommendations in the Guide for the Care and Use of Laboratory Animals of the National Institutes of Health. All procedures were performed in accordance with protocols approved by the Institutional Animal Care and Use Committee (IACUC) of Pennsylvania State University (protocol # 201042827).

### Decision letter and Author response
Decision letter https://doi.org/10.7554/eLife.62071.sa1
Author response https://doi.org/10.7554/eLife.62071.sa2

## Additional files

### Supplementary files
• Supplementary file 1. Supplemental tables.
• Transparent reporting form

### Data availability
Source data and code for generation of all figures can be found here: Code repository location: https://github.com/DrewLab/Turner_Gheres_Proctor_Drew_eLife2020 (copy archived at https://archive.softwareheritage.org/swh:1:rev:a8318fc5cb29b88504fe72f5d3d80867bd9791f2/) Data repository location: https://doi.org/10.5061/dryad.6hdr7sqz5.

The following dataset was generated:

| Author(s) | Year | Dataset title | Dataset URL | Database and Identifier |
|---|---|---|---|---|
| Drew P, Turner K, Gheres K, Proctor E | 2020 | Neurovascular coupling and bilateral connectivity during NREM and REM sleep | https://doi.org/10.5061/dryad.6hdr7sqz5 | Dryad Digital Repository, 10.5061/dryad.6hdr7sqz5 |

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
