## [Decision Letter]

**Acceptance summary:**

This paper combines state-of-the-art techniques to reveal how vascular dynamics and neurovascular coupling are modulated across arousal states. During sleep, cerebral blood flow and its correlation with neuronal activity increase at much higher levels than during wakefulness and sensory stimulation. This study stresses out the importance of carefully monitoring brain states when measuring cerebral blood flow in relation to behaviour and sensory processing.

**Decision letter after peer review:**

Thank you for submitting your article "Neurovascular coupling and bilateral connectivity during NREM and REM sleep" for consideration by *eLife*. Your article has been reviewed by three peer reviewers, and the evaluation has been overseen by a Reviewing Editor and Laura Colgin as the Senior Editor. The following individual involved in review of your submission has agreed to reveal their identity: Aniruddha Das (Reviewer #2).

The reviewers have discussed the reviews with one another and the Reviewing Editor has drafted this decision to help you prepare a revised submission.

The present study reports important findings which shed new light on the physiological basis of brain hemodynamic response. The authors address how hemodynamic measurements are affected by arousal states. Specifically, they show that sleep cycles moving between wakefulness and different sleep stages lead to changes in hemodynamic measurements that are much higher, sometime many-fold, relative to the changes driven by sensory stimulation. These results will be of great interest for researchers in the field of sleep and arousal, neurovascular coupling, fMRI, and the large community of systems neuroscientists performing head-fixed experiments. All three reviewers have expressed enthusiasm about the quality of the study and have, overall, only minor comments regarding the manuscript.

1) Mouse NREM sleep is usually assessed via a more global measure, e.g. EEG, rather than stereotrodes. The use of LFP based measures opens the possibility of detecting local arousal states in awake animals (e.g. Vyazovskiy et al., 2011). The authors should comment on how this may affect their results.

2) The authors should justify why different segment durations were used to quantify wake, NREM, and REM hemodynamics (10 s, 30 s, 60 s) in the "Cortical hemodynamic signals increase during NREM and REM sleep."

3) Figure 6 lacks accompanying statistics in the text.

4) The authors should clarify how they computed average LFP power in Figure 7A of LFP power.

5) Some of the statements about the weakness of neurovascular coupling in the awake state seem overstated, as there can be robust local coupling in the awake state that is not spatially coherent enough to lead to an easily detected correlation with a relatively macroscopic signal such as the 1mm (HbT). The authors should perhaps tone down these statements or justify more clearly their conclusion.

6) The sentence in subsection “Sleep drives larger fluctuations than awake behaviors” regarding hemodynamic measurements should be clarified.

7) Previous works on the topic should be properly cited and discussed.

---

## [Author Response]

The present study reports important findings which shed new light on the physiological basis of brain hemodynamic response. The authors address how hemodynamic measurements are affected by arousal states. Specifically, they show that sleep cycles moving between wakefulness and different sleep stages lead to changes in hemodynamic measurements that are much higher, sometime many-fold, relative to the changes driven by sensory stimulation. These results will be of great interest for researchers in the field of sleep and arousal, neurovascular coupling, fMRI, and the large community of systems neuroscientists performing head-fixed experiments. All three reviewers have expressed enthusiasm about the quality of the study and have, overall, only minor comments regarding the manuscript. Some of these comments are summarized below, more detailed comments can be found in the reviewers' details reports.1) Mouse NREM sleep is usually assessed via a more global measure, e.g. EEG, rather than stereotrodes. The use of LFP based measures opens the possibility of detecting local arousal states in awake animals (e.g. Vyazovskiy et al., 2011). The authors should comment on how this may affect their results.

Discussion and citation added – see Discussion, paragraph two.

2) The authors should justify why different segment durations were used to quantify wake, NREM, and REM hemodynamics (10 s, 30 s, 60 s) in the "Cortical hemodynamic signals increase during NREM and REM sleep."

Further explanation added in the first paragraph of the Results section.

3) Figure 6 lacks accompanying statistics in the text.

Stats for Figure 6 peak and time-to-peak added for MUA/γ-band power of LFP. We have also added a new figure supplement to Figure 6 with the statistics.

4) The authors should clarify how they computed average LFP power in Figure 7A of LFP power.

Clarification added – see subsection “Power spectra and coherence of neural and hemodynamic signals” in the Materials and methods.

5) Some of the statements about the weakness of neurovascular coupling in the awake state seem overstated, as there can be robust local coupling in the awake state that is not spatially coherent enough to lead to an easily detected correlation with a relatively macroscopic signal such as the 1mm (HbT). The authors should perhaps tone down these statements or justify more clearly their conclusion.

We have clarified this in the Discussion, paragraph two.

6) The sentence in subsection “Sleep drives larger fluctuations than awake behaviors” regarding hemodynamic measurements should be clarified.

Added clarification regarding hemodynamic measures and ROI placement with respect to Winder et al., 2017.

7) Previous works on the topic should be properly cited and discussed.

Added comment and citation for Funk et al., 2016, as well as additional clarification with respect to Bergel et al., 2018 – see Introduction paragraph two and Discussion paragraph two.